# FedProx-based Heterogeneity-aware Parameter-free Federated Learning

## Abstract

We propose parameter-free Federated Learning (FL) algorithms based on FedProx. Learning rate-free optimization has been studied in single-node settings, with DoG and its extension DoWG exhibiting strong theoretical and empirical performance. To exploit their success in multi-node FL, we leverage a key insight: a structural similarity between the lemmas for convergence analyses of DoG/DoWG and those of the proximal point algorithm that underlies FedProx. Based on this, we propose two novel FedProx-based algorithms–FedProxLoD and FedProxWLoD–which adaptively determine the proximal weight, serving as FL analogues of DoG and DoWG. We demonstrate tight heterogeneity-aware convergence rates for parameter-free FL that explicitly reflect the impact of data heterogeneity across clients, and show that the proposed algorithms can outperform DoG and DoWG as heterogeneity decreases. Through large-scale numerical experiments on both convex and non-convex models, we validate the effectiveness of the proposed methods. Notably, FedProxWLoD achieved competitive performance with pre-tuned beseline algorithms under moderate data heterogeneity settings.

## 1 Introduction

For advanced collective intelligence involving image, speech, and natural language processing, training large-scale models using extensive computational resources is essential. Federated Learning (FL) McMahan et al. (2017); Kairouz et al. (2021); Konečný et al. (2016); Konečný (2016) is a promising approach for this purpose, as it enables the utilization of distributed computational resources. However, maximizing learning performance in FL typically requires careful pre-tuning of parameters, such as the learning rate. This tuning process is time-consuming and computationally expensive, especially when training large-scale models. Therefore, the key objective is to develop parameter-free FL algorithms that eliminate the need for such manual tuning.

Limited to single-node model training, various learning-rate-free optimization algorithms have been explored. Notable examples include line search methods Nesterov (2015); Grimmer (2024), Polyak stepsize Polyak (1987); Hazan & Kakade (2019); Takezawa et al. (2025), Stochastic Polyak Stepsize (SPS) Loizou et al. (2021), coin betting with normalization Orabona & Pál (2016); Orabona & Cutkosky (2020); Orabona (2023), bisection search Carmon & Hinder (2022), D-Adaptation Defazio & Mishchenko (2023), Distance over Gradient (DoG) Ivgi et al. (2023), and its weighted gradient extension, DoWG Khaled et al. (2023). Among these, DoG and DoWG are particularly promising, providing theoretical convergence analysis for convex loss functions. Despite this theoretical limitation, both methods demonstrate strong empirical performance across a variety of benchmarks, encompassing both convex and non-convex deep learning models. Notably, DoWG addresses early-iteration instability in DoG, leading to improved overall performance.

In contrast, for multi-node FL settings, adaptive gradient methods such as momentum Qian (1999), AdaGrad Duchi et al. (2011); Levy et al. (2018); Ene et al. (2021), Adam Kingma & Ba (2014); Li et al. (2023), have been widely adopted in FL, including FedAvgM Hsu et al. (2019), FedOpt Reddi et al. (2020), and FedAMS Wang et al. (2022). However, these algorithms require careful parameter tuning, most notably the learning rate. FedSPS Mukherjee et al. (2023) is arguably the first attempt to incorporate a learning-rate-free optimizer (SPS) into FL. Nevertheless, it still demands tuning of auxiliary parameters, such as the normalization coefficient and threshold used in learning rate estimation. Furthermore, it lacks a tight heterogeneity-aware convergence rate that explicitly captures the impact of data heterogeneity among clients. This limitation also applies to more recent

Table 1: Comparison of algorithms. The second column categorizes each algorithm based on whether it is parameter-free, applicable to FL settings, and heterogeneity-aware convergence rate. The third column presents the number of iterations to achieve an $\epsilon$-accurate solution for general convex functions. For our algorithms, assumptions of $G$-Lipschitz function, $\zeta$-data heterogeneity, and interpolation condition are given in Assumptions 4, 5, and 6, respectively. Note that $\zeta \leq \mathcal{O}(G)$ always holds.

| Algorithms | Parameter-free | FL | Heterogeneity-aware rate | Convergence rate for general convex function*1 | G-Lipschitz function | L-smooth*2 function | Data heterogeneity | Interpolation condition |
|---|---|---|---|---|---|---|---|---|
| Gradient descent Bubeck et al. (2015) | – | – | – | $\widetilde{\mathcal{O}}(\frac{G^2 D_0^2}{\epsilon^2})$ | ✓ | – | – | – |
| DoG Ivgi et al. (2023) | ✓ | – | – | $\widetilde{\mathcal{O}}(\frac{G^2 D_0^2}{\epsilon^2})$ | ✓ | – | – | – |
| DoWG Khaled et al. (2023) | ✓ | – | – | $\widetilde{\mathcal{O}}(\frac{G^2 D_0^2}{\epsilon^2})$ | ✓ | – | – | – |
| FedAvg Karimireddy et al. (2020) | – | ✓ | ✓ | $\widetilde{\mathcal{O}}\left(\frac{\sqrt{L}\zeta}{\zeta^{\frac{3}{2}}} + \frac{LB^2}{\epsilon}\right)$ | – | ✓ | ✓$((\zeta, B)$-BGD)*3 | – |
| FedProx Li et al. (2020) | – | ✓ | ✓ | $\mathcal{O}(\frac{LB^2}{\epsilon})$ | – | ✓ | ✓$((0, B)$-BGD)*3 | – |
| **FedProxLoD** (this work) | ✓ | ✓ | ✓ | $\widetilde{\mathcal{O}}(\frac{G\zeta D_0^2}{\epsilon^2})$ | ✓ | – | ✓(Assump.5) | ✓ |
| **FedProxWLoD** (this work) | ✓ | ✓ | ✓ | $\widetilde{\mathcal{O}}(\frac{G\zeta D_0^2}{\epsilon^2})$ | ✓ | – | ✓(Assump.5) | ✓ |

*1: $D_0 = \|\mathbf{x}^{(0)} - \mathbf{x}^*\|$: Distance from initial point. *2: $\|\nabla f_i(\mathbf{a}) - \nabla f_i(\mathbf{b})\| \leq L\|\mathbf{a} - \mathbf{b}\|$ for every $\mathbf{a}, \mathbf{b} \in \mathbb{R}^m$. *3: $(\zeta, B)$-BGD requires $(1/n)\sum_{i=1}^n \|\nabla f_i(x)\|^2 \leq \zeta^2 + B^2\|\nabla f(x)\|^2$. Assumption 5 implies $(\sqrt{2}\zeta, \sqrt{2})$-BGD (see Appendix C.2).

work, PAdaMFed Yan et al. (2025). Thus, heterogeneity-aware and fully parameter-free FL remains a largely unexplored and promising research area.

In this paper, we propose FedProxLoD and FedProxWLoD, two heterogeneity-aware, fully parameter-free FL algorithms. Motivated by the theoretical and empirical success of DoG and DoWG, we identify a key insight: a structural similarity between the lemmas used in convergence analysis of DoG/DoWG (Lemma 1 in Section 3) and that for the classical proximal point algorithm Rockafellar (1976); Güler (1991), which underlies FedProx Li et al. (2020; 2024). This connection naturally motivates us to extend FedProx into a fully parameter-free FL by leveraging formulations and theoretical foundations in DoG and DoWG. Our key contributions are summarized as follows:

**FedProx-based heterogeneity-aware parameter-free FL algorithms (in Section 4).** We first reformulate FedProx to incorporate an adaptive proximal weight. In the convergence analysis of this formulation, we derive an inequality (Lemma 2) that closely resembles the one used in the convergence analyses of DoG and DoWG (Lemma 1). We then incorporate the loss difference between the central server and clients, expecting heterogeneity-aware convergence rates. By leveraging structural similarities between the lemmas, we can derive an adaptive determination of the proximal weight. Specifically, the proximal weight is adaptively computed based on the (Weighted) Loss difference over Distance (LoD/WLoD), as formulated in (6). Thus, we refer to our proposed algorithms as FedProxLoD and FedProxWLoD.

**Heterogeneity-aware convergence rates for parameter-free FL (in Section 4).** We provide a convergence analysis of the proposed algorithms under the following assumptions: (i) the local loss function is convex and $G$-Lipschitz (Assumption 4), (iii) the variation in local gradients due to data heterogeneity is bounded, characterized by $\zeta$ (Assumption 5), and (iv) interpolation condition. The main results are presented in Theorem 2, with the corresponding convergence rates summarized in Table 1. Our analysis yields tight, heterogeneity-aware convergence rates, which demonstrate potential improvements over DoG and DoWG as data heterogeneity among clients decreases (i.e., $\zeta \to 0$). Although convergence rates of many (non-parameter-free) FL algorithms have been derived under $L$-smoothness assumption Karimireddy et al. (2020); Woodworth et al. (2020), our analysis does not, implying that it accommodates (non-smooth) general convex objectives.

**Large-scale empirical validations (in Section 5).** We conducted large-scale validations on both convex and non-convex (deep learning) models across image classification and natural language processing tasks. In both scenarios, the proposed FedProxWLoD achieved competitive performance compared to SCAFFOLD Karimireddy et al. (2020) with pre-tuned parameters. These results demonstrate that the potential of FedProx is realized through our fully parameter-free approach.

## 2 SETUPS

The problem is formulated as the minimization of a sum of local loss functions $f_i$, each corresponding to a client $i \in [n]$, as $\min_{\mathbf{x} \in \mathcal{X}} \{f(\mathbf{x}) := \frac{1}{n}\sum_{i=1}^n f_i(\mathbf{x})\}$, where the assumptions used for our convergence analysis in Section 4 are summarized below:

**Assumption 1** (Convex). *A function $f_i$ is twice differentiable and convex for every $i \in [n]$.*

**Assumption 2** (Closed convex). *Domain $\mathcal{X} \subset \mathbb{R}^m$ is a closed convex set.*

**Assumption 3** (Global minimizer). *There exists a minimizer $\mathbf{x}^* \in \mathcal{X}$ of $f$.*

**Assumption 4** ($G$-Lipschitz function). *$f_i$ is $G$-Lipschitz; namely, there exists $G(>0)$ such that $|f_i(\mathbf{a}) - f_i(\mathbf{b})| \leq G\|\mathbf{a} - \mathbf{b}\|$ holds for every $\mathbf{a}, \mathbf{b} \in \mathbb{R}^m$ and $i \in [n]$.*

**Assumption 5** ($\zeta$-bounded data heterogeneity). *There is a constant $\zeta(>0)$ such that holds $\|\nabla f_i(\mathbf{x}) - \nabla f_j(\mathbf{x})\| \leq \zeta$ for every $\mathbf{x} \in \mathbb{R}^m$ and $i, j \in [n]$.*

**Assumption 6** (Interpolation condition). *$\mathbf{x}^*$ is minimizer of $f_i$ for every $i \in [n]$ under Assumption 3.*

**Remark 1.** Assumptions 1–4 are well-used in the optimization field. Under Assumption 4, $\zeta$ in Assumption 5 satisfies $\zeta \leq 2G$ since $\|\nabla f_i(\mathbf{x}) - \nabla f_j(\mathbf{x})\| \leq \|\nabla f_i(\mathbf{x})\| + \|\nabla f_j(\mathbf{x})\| \leq 2G$. Furthermore, we expect that $\zeta \ll G$ in FL scenarios with low to moderate heterogeneity. In particular, when the distributed datasets are i.i.d., we have $\zeta \approx 0$.

**Remark 2.** Relationship between our used assumption and $(\zeta, B)$-Bounded Gradient Dissimilarity (BGD) in Karimireddy et al. (2020) is discussed. The local objectives $\{f_i\}_{i=1}^n$ satisfy $(\zeta, B)$-BGD if $\frac{1}{n}\sum_{i=1}^n \|\nabla f_i(\mathbf{x})\|^2 = \zeta^2 + B^2\|\nabla f(\mathbf{x})\|^2$, $\forall \mathbf{x}, f = \frac{1}{n}\sum_i^n f_i$. If Assumption 5 holds, then $(\sqrt{2}\zeta, \sqrt{2})$-BGD holds (see Appendix C.2).

**Remark 3.** Assumption 6 is introduced solely to ensure the monotonic decrease of the quantity $\|\mathbf{x}_i^{t+1} - \mathbf{x}^*\|^2$. This assumption naturally holds in interpolation regimes, such as in (approximated) kernel methods, overparameterized deep neural networks, where the training loss is minimized to zero Jacot et al. (2018); Allen-Zhu et al. (2019). Moreover, the combination of Assumptions 5 and 6 is not redundant in the general convex settings, as local client minimizers may not be unique; without collaboration, the resulting solutions may differ across $n$ clients.

## 3 PRELIMINARIES

Section 3.1 introduces a brief overview of DoG and DoWG, learning-rate-free optimization algorithms for single-node settings. Section 3.2 provides FedProx as a baseline FL algorithm.

### 3.1 DoG AND DoWG: LEARNING-RATE-FREE OPTIMIZATION WITHIN SINGLE NODE

As discussed in Section 1, several approaches have been proposed for learning-rate-free optimization in the single-node setting. Among them, DoG Ivgi et al. (2023) and its extension DoWG Khaled et al. (2023) demonstrated notable theoretical and empirical success. In this context, the projected gradient descent update for the parameter $\mathbf{x} \in \mathcal{X}$, using an adaptive learning rate $\eta^{(t)}(>0)$, is given by

$$\mathbf{x}^{(t+1)} = \Pi_{\mathcal{X}}\big(\mathbf{x}^{(t)} - \eta^{(t)}\nabla f(\mathbf{x}^{(t)})\big), \tag{1}$$

where $f$ denotes loss function and $\Pi_{\mathcal{X}}$ is projection on domain $\mathcal{X}$.

In DoG and DoWG, the learning rate is given by Distance over (Weighted) Gradient; namely, it is determined by the (maximum) distance between the initial point and the observed iteration $t$, denoted by $r^{(t)} = \max_{k \leq t}\|\mathbf{x}^{(t)} - \mathbf{x}^{(0)}\|$ over (running sum of) gradient norm as follow:

$$[\textbf{DoG}] \ \eta^{(t)} = \frac{r^{(t)}}{\sqrt{\sum_{k \leq t}\big\|\nabla f(\mathbf{x}^{(k)})\big\|^2}}, \quad [\textbf{DoWG}] \ \eta^{(t)} = \frac{(r^{(t)})^2}{\sqrt{\sum_{k \leq t}(r^{(k)})^2\big\|\nabla f(\mathbf{x}^{(k)})\big\|^2}}. \tag{2}$$

These adaptive learning rates are motivated by the following inequality, which is commonly used in convergence analysis in DoG and DoWG.

**Lemma 1** (Ivgi et al. (2023)). *Suppose that $f$ is convex, and has minimizer $\mathbf{x}^* \in \mathcal{X}$. Define $d^{(t)} = \|\mathbf{x}^{(t)} - \mathbf{x}^*\|^2$. For the iterations generated by (1), we have*

$$f(\mathbf{x}^{(t+1)}) - f(\mathbf{x}^*) \leq \frac{1}{2\eta^{(t)}}\Big(\big(d^{(t)}\big)^2 - \big(d^{(t+1)}\big)^2\Big) + \frac{\eta^{(t)}}{2}\big\|\nabla f(\mathbf{x}^{(t)})\big\|^2.$$

In Lemma 1, the RHS consists of two terms: the one-step distance $(d^{(t)})^2 - (d^{(t+1)})^2$ and the gradient norm $\|\nabla f(\mathbf{x}^{(t)})\|^2$. As detailed in Appendix A, a weighted summation of Lemma 1 yields a convergence rate. While the specific weighted strategies differ between DoG and DoWG, their convergence rate orders remain the same, as shown below.

**Theorem 1** (Convergence rates of DoG Ivgi et al. (2023) and DoWG Khaled et al. (2023) [1] ). *Suppose that $f$ is convex, has minimizer $\mathbf{x}^* \in \mathcal{X}$, and $G$-Lipschitz. Let $\{\mathbf{x}^{(t)}\}_{t=0}^T$ be the iterates generated by (1) using adaptive learning rates in (2), we have:*

$$f(\bar{\mathbf{x}}^{(T)}) - f(\mathbf{x}^*) \leq \widetilde{\mathcal{O}}\Big(\frac{GD_0}{\sqrt{T}}\Big),$$

*where $\bar{\mathbf{x}}^{(T)} = \frac{1}{T} \sum_{t=0}^{T-1} \mathbf{x}^{(t)}$ and $D_0 = \|\mathbf{x}^{(0)} - \mathbf{x}^*\|$.*

Although DoG and DoWG share the same theoretical convergence rates, several numerical experiments in Khaled et al. (2023) demonstrated the practical advantages of the adaptive weighting scheme employed in DoWG in (2). This benefit is likely because the adaptively weighted scheme in DoWG helps mitigate the unstable behavior that often occurs around early iterations.

### 3.2 FEDPROX: A BASELINE INCORPORATING FL-SPECIFIC PARAMETERS TO BE PRE-TUNED

As outlined in Section 1, we adopt FedProx as our baseline FL algorithm. Let $\mathbf{x} \in \mathcal{X}$ denote the global parameter maintained by the central server, and let $\mathbf{x}_i \in \mathcal{X}$ represent the local parameter on client $i \in [n]$. The objective is to find a global parameter $\mathbf{x}$ that minimizes the global loss function defined as $f(\mathbf{x}) = \frac{1}{n} \sum_{i=1}^n f_i(\mathbf{x})$, where each $f_i$ is the local loss function associated with client $i$, which may hold a statistically-biased local dataset $\mathcal{D}_i$. FedProx proceeds by iteratively alternating between: i) local parameter updates on each client, and ii) mixing of the local parameters on the central server:

$$[\textbf{Client } i] \; \mathbf{x}_i^{(t+1)} = \arg\min_{\mathbf{y} \in \mathcal{X}} \Big( f_i(\mathbf{y}) + \frac{\mu}{2} \left\| \mathbf{y} - \mathbf{x}^{(t)} \right\|^2 \Big), \; [\textbf{Server}] \; \mathbf{x}^{(t+1)} = \frac{1}{n} \sum_{i=1}^n \mathbf{x}_i^{(t+1)}, \quad (3)$$

where $\mathbf{x}^{(0)}$ is given, and $\mu \, (> 0)$ is the proximal weight that must be pre-tuned. Since $\mu$ is an FL-specific parameter that controls the proximity between global and local parameters, improper tuning can adversely affect the performance of FedProx.

To solve the subproblem in (3), each client updates its local parameter such that $\mathbf{x}_i^{(t+1)}$ satisfies

$$\nabla f_i(\mathbf{x}_i^{(t+1)}) + \mu(\mathbf{x}_i^{(t+1)} - \mathbf{x}^{(t)}) \in -\partial \Pi_{\mathcal{X}}(\mathbf{x}_i^{(t+1)}), \quad (4)$$

where subdifferential operator $\partial$ is used as $\Pi_{\mathcal{X}}$ can be interpreted as a function including non-differentiable points:

$$\Pi_{\mathcal{X}}(\mathbf{x}) = \begin{cases} 0 & (\mathbf{x} \in \mathcal{X}) \\ \infty & (\text{otherwise}) \end{cases}. \quad (5)$$

Since (4) is generally intractable in closed form for complex loss functions (e.g., deep learning models), FedProx implementations typically approximate it by performing multiple iterations of the following update: $\mathbf{x}_i^{(t+1)} = \Pi_{\mathcal{X}}(\mathbf{x}_i^{(t)} - \eta(\nabla f_i(\mathbf{x}_i^{(t)}) + \mu(\mathbf{x}_i^{(t)} - \mathbf{x}^{(t)})))$, which can be interpreted as solving a quadratic approximation of $f_i$ around the current iterate $\mathbf{x}_i^{(t)}$ (see Appendix B). In practice, FedProx requires careful tuning of hyperparameters–most notably the proximal weight $\mu$ (and learning rate $\eta$, when using the approximated local update rule)–to achieve good performance. While Appendix E provides empirical evidence underscoring the importance of this tuning, the reliance on such pre-specified hyperparameters highlights the need for parameter-free FL algorithms that remove time-consuming manual tuning.

## 4 PROPOSED ALGORITHMS

In this section, we propose FedProxLoD and FedProxWLoD–parameter-free FL algorithms built upon FedProx–and present their convergence analysis.

---

[1] Since their analysis strategies differ (DoG provides high-probability bounds, whereas DoWG also presents deterministic bounds), we include Appendix A to derive deterministic bounds for both DoG and DoWG.

---

**Algorithm 1** Proposed Algorithms (FedProxLoD and FedProxWLoD)

---

1: Initialization $\mathbf{x}^{(0)} = \mathbf{x}_{\text{out}}^{(0)} = \mathbf{x}_{\text{best}}^{(0)}, r^{(0)}(> 0), u^{(0)}(> 0), w_2^{(0)} = 0$

2: **if** (FedProxLoD) $\mu^{(0)} = r^{(0)}/\sqrt{u^{(0)}}$, **else if** (FedProxWLoD) $\mu^{(0)} = (r^{(0)})^2/\sqrt{u^{(0)}}$ **end**

3: **for** $t = 0, 1, \ldots, T-1$ **do**

4:    ▷ Client procedure

5:    **for** $i = 1, \ldots, n$ **do**

6:       $\mathbf{x}_i^{(t+1)} = \arg\min_{\mathbf{y} \in \mathcal{X}} \big(f_i(\mathbf{y}) + \frac{\mu^{(t)}}{2}\|\mathbf{y} - \mathbf{x}_{\text{best}}^{(t)}\|^2\big)$

7:    **end for**

8:    $\textbf{Transmit}_{\text{Client } i \to \text{Server}}(\mathbf{x}_i^{(t+1)}, f_i(\mathbf{x}_i^{(t+1)}))$

9:    ▷ Server procedure

10:    $\mathbf{x}^{(t+1)} = \frac{1}{n}\sum_{i=1}^n \mathbf{x}_i^{(t+1)}$

11:    $r^{(t+1)} = \max\{\|\mathbf{x}^{(t+1)} - \mathbf{x}^{(0)}\|, r^{(t)}\}$

12:    $\Delta^{(t+1)} = \mu^{(t)} \cdot \max\{f(\mathbf{x}^{(t+1)}) - \frac{1}{n}\sum_{i=1}^n f_i(\mathbf{x}_i^{(t+1)}) - \frac{\mu^{(t)}}{2n}\sum_{i=1}^n \|\mathbf{x}_i^{(t+1)} - \mathbf{x}^{(t)}\|^2, 0\}$

13:    **if** (FedProxLoD) **then**

14:       $u^{(t+1)} = u^{(t)} + \Delta^{(r+1)}$

15:       $\mu^{(t+1)} = \sqrt{u^{(t+1)}}/r^{(t+1)}$

16:       $w_1^{(t+1)} = \min\{\frac{\mu^{(t+1)}}{\mu^{(t)}}, 1\} r^{(t+1)}$

17:    **else if** (FedProxWLoD) **then**

18:       $u^{(t+1)} = u^{(t)} + (r^{(t+1)})^2 \Delta^{(r+1)}$

19:       $\mu^{(t+1)} = \sqrt{u^{(t+1)}}/(r^{(t+1)})^2$

20:       $w_1^{(t+1)} = \min\{\frac{\mu^{(t+1)}}{\mu^{(t)}}, 1\}(r^{(t+1)})^2$

21:    **end if**

22:    $w_2^{(t+1)} = w_2^{(t)} + w_1^{(t+1)}$

23:    $\mathbf{x}_{\text{out}}^{(t+1)} = \frac{1}{w_2^{(t+1)}}\big(w_2^{(t)}\mathbf{x}_{\text{out}}^{(t)} + w_1^{(t+1)}\mathbf{x}^{(t+1)}\big)$

24:    $\mathbf{x}_{\text{best}}^{(t+1)} = \arg\min_{\mathbf{x} \in \{\mathbf{x}_{\text{out}}^{(t+1)}, \mathbf{x}_{\text{best}}^{(t)}\}} f(\mathbf{x})$

25:    $\textbf{Transmit}_{\text{Server} \to n \text{ clients}}(\mathbf{x}_{\text{best}}^{(t+1)}, \mu^{(t+1)})$

26: **end for**

---

**Main idea.** First, we begin by extending the FedProx procedure in (3) to allow the proximal weight to vary adaptively across iterations $t$. Specifically, the client update in (3) is replaced with $\mathbf{x}_i^{(t+1)} = \arg\min_{\mathbf{y} \in \mathcal{X}}(f_i(\mathbf{y}) + \frac{\mu^{(t)}}{2}\|\mathbf{y} - \mathbf{x}^{(t)}\|^2)$, where the adaptive proximal weight $\mu^{(t)}$ is dynamically determined on the central server and broadcast to all $n$ clients.

To determine $\mu^{(t)}$ based on observable quantities (e.g., gradient norm or training loss), we derive a key inequality analogous to that in Lemma 1. This inequality is inspired by the convergence analysis of the proximal point algorithm Rockafellar (1976); Güler (1991), which underlies FedProx and exhibits a structural similarity to Lemma 1. While detailed derivations are presented in Appendix C, following Lemma 2 is obtained based on two core properties: i) the convexity of $f_i$, which yields as $f_i(\mathbf{x}_i^{(t+1)}) - f_i(\mathbf{x}^*) \le \langle \nabla f_i(\mathbf{x}_i^{(t+1)}), \mathbf{x}_i^{(t+1)} - \mathbf{x}^* \rangle$; ii) gradient bound derived from local parameter update rule in (4), given by $\nabla f_i(\mathbf{x}_i^{(t+1)}) \in -\mu^{(t)}(\mathbf{x}_i^{(t+1)} - \mathbf{x}^{(t)}) - \partial \Pi_{\mathcal{X}}(\mathbf{x}_i^{(t+1)})$.

**Lemma 2.** *Suppose that Assumptions 1-3 hold. For the iterations generated by (3) employing adaptive proximal weight $\mu^{(t)}$, we have*

$$f(\mathbf{x}^{(t+1)}) - f(\mathbf{x}^*) \le \frac{\mu^{(t)}}{2}\Big((d^{(t)})^2 - (d^{(t+1)})^2\Big) + \frac{1}{\mu^{(t)}}\Delta^{(t+1)},$$

*where $\Delta^{(t+1)} = \mu^{(t)} \cdot \max\{f(\mathbf{x}^{(t+1)}) - \frac{1}{n}\sum_{i=1}^n f_i(\mathbf{x}_i^{(t+1)}) - \frac{\mu^{(t)}}{2n}\sum_{i=1}^n \|\mathbf{x}_i^{(t+1)} - \mathbf{x}^{(t)}\|^2, 0\}$, and we call this loss difference.*

In Lemma 2, we introduce the loss difference $\Delta$, which enables heterogeneity-aware convergence rates derived later in this section. Comparing Lemmas 1 and 2, we observe a clear structural similarity, aside from the use of the loss difference in place of the gradient norm and the substitution of the

proximal weight for the learning rate. This observation naturally motivates an adaptive determination of the proximal weight. Analogous to the learning rate formulation in DoG/DoWG, the proximal weight is expressed as the (Weighted) Loss difference over Distance, referred to as LoD/WLoD, as

$$[\textbf{FedProxLoD}] \ \ \mu^{(t)} = \frac{\sqrt{\sum_{k \le t} \Delta^{(k)}}}{r^{(t)}}, \quad [\textbf{FedProxWLoD}] \ \ \mu^{(t)} = \frac{\sqrt{\sum_{k \le t} (r^{(k)})^2 \Delta^{(k)}}}{(r^{(t)})^2}. \quad (6)$$

**Algorithm construction.** Based on the FedProx (3) with adaptive proximal weights in (6), we construct the FedProxLoD and FedProxWLoD, as detailed in Algorithm 1. The core procedures–local model updates on each client and mixing on the central server in (3)–are implemented in Lines 6 and 10, respectively. To compute the proximal weight as defined in (6), the maximum distance concerning the global parameter $r^{(t+1)}$, is computed in Line 11, and the loss difference $\Delta^{(t+1)}$ is updated in Line 12. To compute the loss difference $\Delta^{(t+1)}$, both the global loss $f(\mathbf{x}^{(t+1)})$ and the local loss $f_i(\mathbf{x}_i^{(t+1)})$ are required. This involves: i) computing the global loss on the central server, which requires access to a balanced subset of data from all $n$ clients, and ii) transmitting not only $n$ local models but also local loss values computed on the client's training datasets as in Line 8. Using computed distance $r^{(t+1)}$ and loss difference $\Delta^{(t+1)}$, proximal weight $\mu^{(t+1)}$ is updated through Lines 13–21.

However, to establish a rigorous convergence analysis when updating proximal weight $\mu^{(t+1)}$ using distance $r^{(t+1)}$ and loss difference $\Delta^{(t+1)}$, additional model merging on the central server, specified in Lines 22–24, is required. On the RHS of Lemma 2, the loss difference $\Delta^{(t+1)}$ is scaled by $1/\mu^{(t)}$ rather than by $\mu^{(t+1)}$. To update $\mu^{(t+1)}$ consistently with $\Delta^{(t+1)}$, we multiply both sides of Lemma 2 by $\min\{\frac{\mu^{(t)}}{\mu^{(t+1)}}, 1\}$. Summing over iterations with two weighing schemes, following DoG and DoWG, then yields the following:

**Lemma 3.** *Suppose that Assumptions 1-3, 6 hold. For the iterations generated by (3) employing adaptive proximal weight $\mu^{(t)}$, we have*

[**FedProxLoD**]

$$\sum_{t=0}^{T-1} \min\Big\{\frac{\mu^{(t)}}{\mu^{(t+1)}}, 1\Big\} r^{(t)} \Big(f(\mathbf{x}^{(t+1)}) - f(\mathbf{x}^*)\Big) \le \sum_{t=0}^{T-1} \frac{r^{(t)}\mu^{(t)}}{2} \Big((d^{(t)})^2 - (d^{(t+1)})^2\Big) + \sum_{t=0}^{T-1} \frac{r^{(t)}}{\mu^{(t+1)}} \Delta^{(t+1)},$$

[**FedProxWLoD**]

$$\sum_{t=0}^{T-1} \min\Big\{\frac{\mu^{(t)}}{\mu^{(t+1)}}, 1\Big\} (r^{(t)})^2 \Big(f(\mathbf{x}^{(t+1)}) - f(\mathbf{x}^*)\Big) \le \sum_{t=0}^{T-1} \frac{(r^{(t)})^2\mu^{(t)}}{2} \Big((d^{(t)})^2 - (d^{(t+1)})^2\Big) + \sum_{t=0}^{T-1} \frac{(r^{(t)})^2}{\mu^{(t+1)}} \Delta^{(t+1)}.$$

To handle the extraneous coefficient $\min\{\frac{\mu^{(t)}}{\mu^{(t+1)}}, 1\}$ on the LHS of Lemma 3, we introduce the model merging (Lines 22–24). This allows the expression to be recast into a standard form commonly employed in convergence analysis, with the details provided in Appendix C.

**Convergence analysis.** Using Lemma 3, a convergence analysis of FedPRoxLoD/FedPRoxWLoD in Algorithm 1 is provided, with details presented in Appendix C. As shown in Lemma 11 in Appendix C, the loss difference included in the RHS of Lemma 2 can be bounded as $\Delta^{(t+1)} \le \mathcal{O}(\zeta G)$ under certain assumptions. On the other hand, the corresponding term in Lemma 1 can be bounded as $\|\nabla f(\mathbf{x}^{(t)})\|^2 \le \mathcal{O}(G^2)$. This distinction is clearly reflected in the convergence rates presented below.

**Theorem 2** ((Informal)[2] convergence rates of FedProxLoD and FedProxWLoD). *Suppose that Assumptions 1–5 hold. For the iterations generated by Algorithm 1 and a certain large $T$, we have:*

$$f(\mathbf{x}_{best}^{(T)}) - f(\mathbf{x}^*) \le \widetilde{\mathcal{O}}\Big(\frac{\sqrt{\zeta G}D_0}{\sqrt{T}}\Big),$$

*where $\mathbf{x}_{best}^{(T)} := \arg\min_{\mathbf{x} \in \{\mathbf{x}^{(T')}\}_{T' \in [T]}} f(\mathbf{x})$.*

---

[2]Formal statement is given in Appendix C.

Theorem 2 demonstrates tight heterogeneity-aware convergence rates for FedProxLoD and FedProx-WLoD, enabled by the use of loss difference in Lemmas 2 and 3. Both algorithms perform better as data heterogeneity decreases ($\zeta \to 0$). Moreover, $\zeta \leq 2G$ holds under Assumptions 4 and 5, as discussed in Section 2. This implies that the convergence rate in Theorem 2 can be bounded by $\widetilde{\mathcal{O}}(\frac{GD_0}{\sqrt{T}})$, which matches the rate in Theorem 1 for single-node parameter-free optimization. This result highlights the benefit of distributed optimization using multiple clients.

**Implementation techniques.** As empirically used in the original FedProx, the approximated update rule can be applicable in Line 6 in Algorithm 1. Specifically, the update: $\mathbf{x}_i^{(t+1)} = \Pi_{\mathcal{X}}(\mathbf{x}_i^{(t)} - \eta^{(t)}(\nabla f_i(\mathbf{x}_i^{(t)}) + \mu^{(t)}(\mathbf{x}_i^{(t)} - \mathbf{x}^{(t)})))$ can be iteratively applied for multiple steps. Under this approximation, existing learning rate-free optimization methods such as DoG and DoWG can be employed to determine $\eta^{(t)}$. The complete procedure is detailed in Algorithm 3 in Appendix D, and it is used in the numerical experiments in Section 5. It is important to note, however, that the convergence rates shown in Theorem 2 are based on the original formulation of Algorithm 1, and do not consider the potential impact of this approximation.

## 5 EXPERIMENTS

We empirically evaluate the proposed algorithms using image classification and natural language processing tasks. *The main goal is to assess whether the proposed parameter-free FL algorithms can achieve comparable to, or better than, baseline FL algorithms with pre-tuned parameters.* The experimental setups are described in Section 5.1, and the results are presented in Section 5.2.

### 5.1 EXPERIMENTAL SETUPS

**NW configuration and data distribution.** The network configuration consists of a central server and $n = 15$ clients[3]. Communication between the central server and the clients is permitted once every $K = 100$ local parameter updates. To empirically evaluate the impact of data heterogeneity, the training dataset is partitioned across the $n$ clients according to a Dirichlet distribution with parameter $\alpha \in \{1, 0.1\}$ Vogels et al. (2021). Smaller values of $\alpha$ correspond to a strongly imbalanced heterogeneous data setting. Consequently, each client may hold a different number of data samples. The resulting data distributions for each dataset across the $n$ clients are visualized in Figure 1.

**Comparison algorithms.** We compare FL algorithms in the following categories:

**(C1)-(C7) Fundamental FL algorithms requiring parameter pre-tuning.** We evaluate the following algorithms: (C1) FedAvg McMahan et al. (2017), (C2) FedAvg with Momentum (FedAvgM) Hsu et al. (2019), (C3) FedProx Li et al. (2020), (C4) SCAFFOLD Karimireddy et al. (2020), and FedOpt variants–(C5) FedAdaGrad and (C6) FedAdam Reddi et al. (2020), and (C7) FedSPS Mukherjee et al. (2023). All algorithms require parameter pre-tuning. For (C1)-(C6), we employ cosine annealing as the learning rate scheduler, reducing it by a factor of $1/10$ during training. The initial learning rate is selected from $\eta \in \{1, 0.1, 0.01, 0.001\}$. For (C3) FedProx, we additionally tune the proximal weight within $\mu \in \{0.1, 0.01, 0.001\}$. For (C4) SCAFFOLD, we tune the learning rate by setting the global and local learning rates equal. For the learning-rate-free optimization (C7) FedSPS, we tune the normalization coefficient $c \in \{0.1, 1.0\}$.

**(P1)-(P2) Proposed parameter-free algorithms and their ablations (P1')-(P2').** For the proposed algorithms, (P1) FedProxLoD and (P2) FedProxWLoD, we adopt the version in Algorithm 3 in Appendix D, as its local update procedure is suitable for non-convex deep learning models. As their variants as ablations, we empirically examine the effectiveness of extra model merging on the server (Lines 22–24 in Algorithm 1) for each (P1) and (P2).

**Tasks and models.** Theoretical convergence analysis in Section 4 is conducted under the assumption that the loss function is convex. Following prior studies, DoG and DoWG in Section 3.1, we evaluate algorithms on both convex and non-convex models to demonstrate their practical applicability. Our benchmark tests are organized into three categories as follows:

**(T1) Image classification task with convex model.** We evaluate performance on the fashion MNIST (fMNIST) dataset Xiao et al. (2017) using a convex model. Specifically, we utilize a two-layer

---

[3]We also tested using a central server with $n = 7$ clients, as summarized in Appendix E

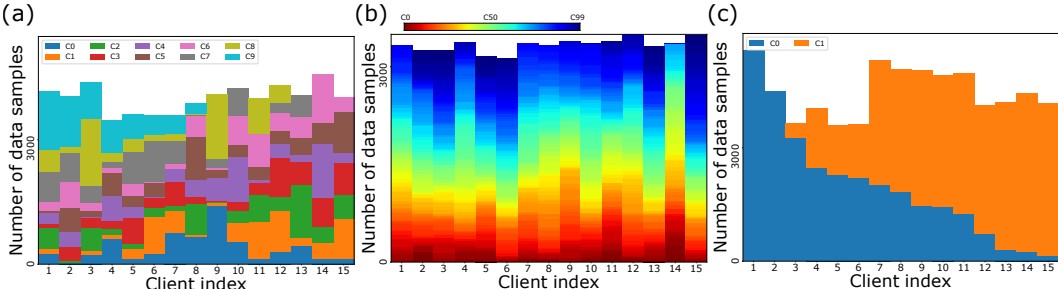

Figure 1: Data distributions using $n = 15$ clients and $\alpha = 1$: (a) fMNIST classification in (T1), (b) CIFAR-100 classification in (T2), and (c) SST-2 classification in (T3).

Table 2: Best test accuracy and test accuracy at last round (last test accuracy) under $n = 15$ and $\alpha = 1$. In the comparing algorithms (C1)-(C7), pre-tuning of parameters was conducted. Despite not requiring parameter pre-tuning, our parameter-free algorithms (P1), (P2), (P1'), (P2') achieved competing performance relative to the best performance of pre-tuned baseline algorithms (C1)-(C7).

| Algorithms | Parameters | (T1) Convex-fMNIST | | (T2) ResNet-18-CIFAR-100 | | (T3) BERT-SST-2 | |
|---|---|---|---|---|---|---|---|
| | | Best test acc. | Last test acc. | Best test acc. | Last test acc. | Best test acc. | Last test acc. |
| (C1) FedAvg McMahan et al. (2017) | $\{1, 0.1, 0.01, 0.001\} \in \eta$ | 0.8837 | 0.8820 | 0.6736 | 0.6658 | 0.9278 | 0.9106 |
| (C2) FedAvgM Hsu et al. (2019) | $\{1, 0.1, 0.01, 0.001\} \in \eta$ | 0.8850 | 0.8825 | 0.6676 | 0.6624 | 0.9289 | 0.9232 |
| (C3) FedProx Li et al. (2020) | $\{ 1, 0.1, 0.01, 0.001 \} \in \eta$ $\{ 0.1, 0.01, 0.001 \} \in \mu$ | 0.8848 | 0.8821 | 0.6717 | 0.6652 | **0.9369** | **0.9323** |
| (C4) SCAFFOLD Karimireddy et al. (2020) | $\{1, 0.1, 0.01, 0.001\} \in \eta$ | **0.8863** | **0.8843** | **0.6877** | **0.6786** | 0.9266 | 0.9083 |
| (C5) FedAdaGrad Reddi et al. (2020) | $\{1, 0.1, 0.01, 0.001\} \in \eta$ | 0.8850 | 0.8827 | 0.6551 | 0.6502 | 0.9220 | 0.9117 |
| (C6) FedAdam Reddi et al. (2020) | $\{1, 0.1, 0.01, 0.001\} \in \eta$ | 0.8835 | 0.8823 | 0.6667 | 0.6596 | 0.9232 | 0.9106 |
| (C7) FedSPS Mukherjee et al. (2023) | $\{1, 0.1\} \in c$ | 0.8829 | 0.8810 | 0.6254 | 0.6241 | 0.9140 | 0.9014 |
| (P1) FedProxLoD | Parameter-free | 0.8782 | 0.8782 | 0.6731 | 0.6729 | 0.8922 | 0.8899 |
| (P2) FedProxWLoD | Parameter-free | 0.8789 | 0.8789 | 0.6788 | 0.6787 | 0.9220 | 0.9220 |
| (P1') (P1) w/o model merge | Parameter-free | 0.8785 | 0.8784 | 0.6688 | 0.6678 | 0.8922 | 0.8899 |
| (P2') (P2) w/o model merge | Parameter-free | **0.8801** | **0.8801** | **0.6807** | **0.6795** | 0.9197 | 0.9197 |

multi-layer perceptron with the weights of the first layer fixed (i.e., untrainable) to ensure convexity, while maintaining a large model size $m$. This model choice is motivated by Assumption 6, which holds in interpolation regimes as discussed in Section 2. For this aim, we set a large hidden layer units $8,192$. We use a batch size of $64$, and examine two levels of data heterogeneity $\alpha \in \{1, 0.1\}$.

**(T2) Image classification task with non-convex model (ResNet-18).** We evaluate performance on the CIFAR-100 datasets Krizhevsky et al. (2009) using a non-convex model, ResNet-18 He et al. (2016). Note that the batch normalization layers were replaced by group normalization layers Wu & He (2018) to account for potential data heterogeneity in local datasets. We use a batch size of $64$, and examine three levels of data heterogeneity, setting $\alpha \in \{1, 0.1\}$.

**(T3) Natural language processing task with non-convex model (BERT).** We evaluate performance on the SST-2 classification Wang et al. (2018) using a non-convex model, BERT Devlin et al. (2018), handling as fine-tuning tasks using a pre-trained model. Experiments are conducted with a batch size of $32$ and two levels of data heterogeneity $\alpha \in \{1, 0.1\}$. To ensure stable convergence, a warm-up interval of $10$ communication rounds for $\eta$ is applied to the comparing algorithms (C1)-(C7).

## 5.2 EXPERIMENTAL RESULTS

Table 2 presents the best test accuracies and test accuracies at last round (last test accuracies) under a moderate data heterogeneity setting ($\alpha = 1$). For the baseline algorithms (C1)-(C7), parameters $(\eta, \mu)$ are pre-tuned to yield their best performance. Results for additional scenario ($n \in \{15, 7\}, \alpha \in \{1, 0.1\}$) are summarized in Appendix E. From Table 2, we observe that SCAFFOLD with pre-tuned parameters demonstrated strong performance among the baseline algorithms. This is expected, as SCAFFOLD incorporates additional control variates into local model updates to address data heterogeneity. Despite being parameter-free, our proposed FedProxWLod showed competitive performance compared to pre-tuned SCAFFOLD. Since FL requires substantial computational resources, achieving strong performance without the need for parameter pre-tuning is a significant advantage. While parameter-free algorithms are typically theoretically analyzed only for convex functions and often fail in training practical deep learning models, it is noteworthy that our proposed algorithms empirically overcome this limitation. This effectiveness can be attributed to the successful exploitation of the analogy with DoG/DoWG. While FedProxLod also demonstrated strong performance, FedProxWLod is preferable, as it consistently outperforms FedProxLod. Furthermore,

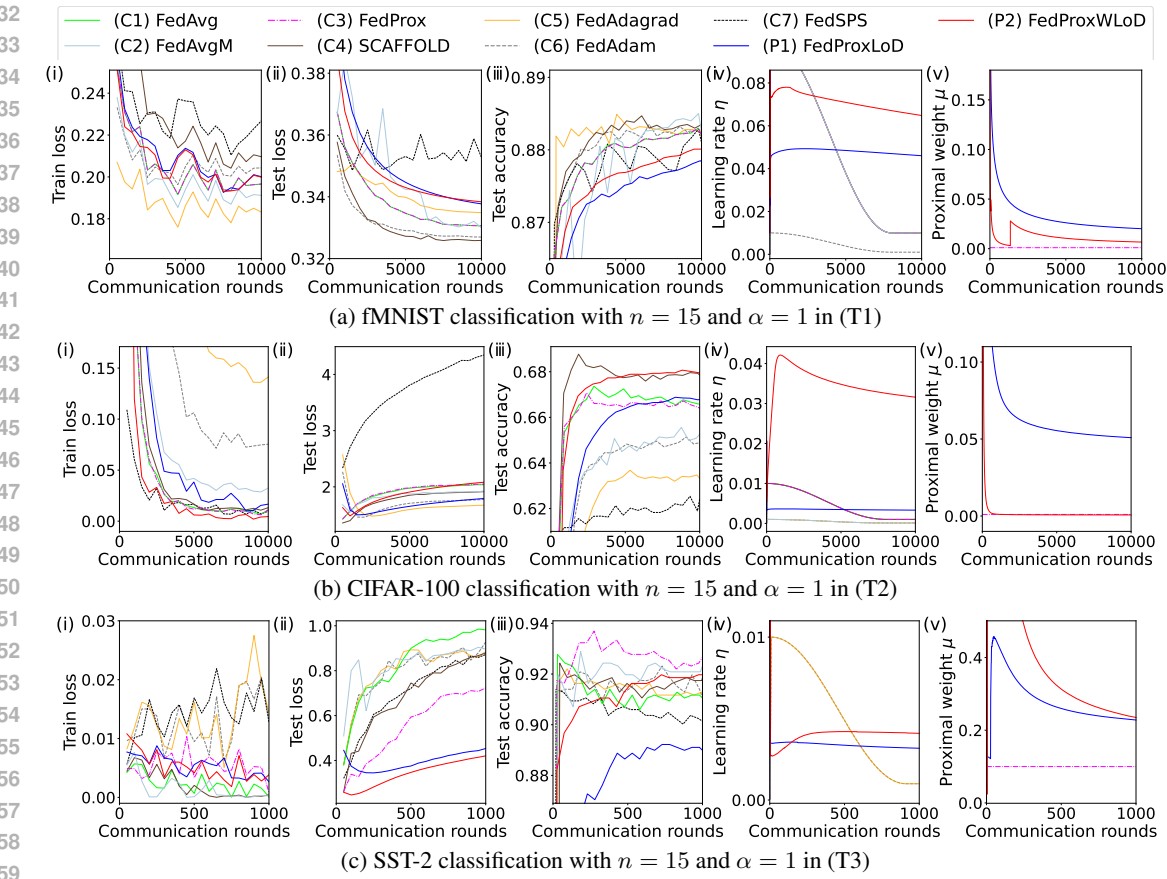

(a) fMNIST classification with $n = 15$ and $\alpha = 1$ in (T1)

(b) CIFAR-100 classification with $n = 15$ and $\alpha = 1$ in (T2)

(c) SST-2 classification with $n = 15$ and $\alpha = 1$ in (T3)

Figure 2: Convergence curves illustrating (i) train loss, (ii) test loss, (iii) test accuracy, and the evolution of (iv) learning rate and (v) proximal weight for a part of the benchmark tests.

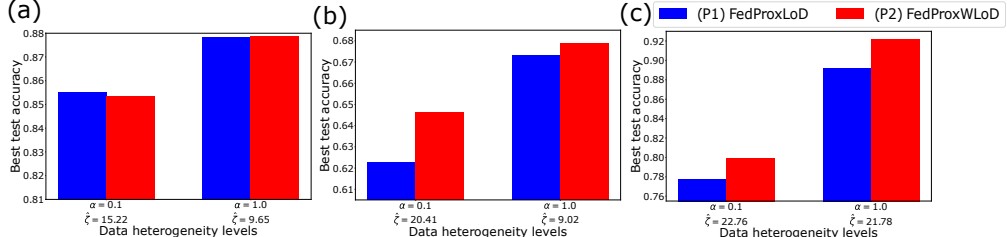

Figure 3: Investigation into the impact of data heterogeneity: (a) fMNIST classification in (T1), (b) CIFAR-100 classification in (T2), and (c) SST-2 classification in (T3).

while enabling extra model merging is necessary for theoretical convergence analysis, it does not appear to be critically important for improving empirical performance. These empirical findings suggest that adaptive and optimal tuning of parameters can fully unlock the potential of FedProx.

Figure 2 presents the convergence curves in terms of train loss, test loss, test accuracy, and the trajectories of the parameters $(\eta, \mu)$. Compared to FedProxLod, FedProxWLod employed a larger learning rate throughout the learning process (this is shown in Khaled et al. (2023)). In contrast, in both FedProxLod and FedProxWLod, the proximal weight gradually decreased to nearly zero over the course of the iterations, since the running sum of the loss difference in (6) did not significantly increase in later iterations. These observations suggest that enforcing stronger model proximity with a larger learning rate during the early iterations plays a key role in the effectiveness of FedProxWLod.

Given the heterogeneity-aware convergence rates in Theorem 2, we investigated the best test accuracy across different heterogeneity levels in Figure 3. For each heterogeneity level ($\alpha$), the heterogeneity parameter $\zeta$ from Assumption 5 was estimated by computing the norm of gradient differences

between two clients over multiple iterations and selecting the maximum observed value. As data heterogeneity increases, the performance of both FedProxLoD and FedProxWLoD degraded. However, FedProxWLoD in particular maintains competitive performance even under strong heterogeneity levels, providing empirical support for the theoretical claims in Theorem 2.

## 6 CONCLUSION

We proposed FedProxLoD and FedProxWLoD as heterogeneity-aware parameter-free FL algorithms. Our key finding lies in the inequality derived for the convergence analysis of FedProx with an adaptive proximal weight (Lemma 2). By exploiting the structural similarity between Lemmas 1 and 2, we construct parameter-free FL algorithms–FedProxLoD and FedProxWLoD–as described in Algorithm 1. Owing to the use of loss difference in adaptive determination of proximal weight (in (6)), the resulting convergence rates under $G$-Lipschitz convex loss functions (Theorem 2) are explicitly heterogeity-aware–that is, desiring tight convergence analysis is achieved. Moreover, we show that the proposed algorithms can outperform DoG and DoWG as data heterogeneity across clients decreases. Through large-scale numerical experiments across both convex and non-convex models, we validated the effectiveness of the proposed algorithms. Notably, FedProxWLoD achieved performance competitive with parameter-tuned SCAFFOLD. This is a surprising result in the FL field, where many parameter-free algorithms often underperform in training practical (non-convex) deep learning models.

**Ethics statement**  Our work does not involve human subjects, dataset release practices, or related issues; therefore, we identify no ethical concerns.

**Reproducibility statement**  For reproducibility, we provide both appendix sections and source code as supplementary material. For the theoretical results (e.g., Theorems 1 and 2), the corresponding proofs are given in Appendix A and Appendix C, respectively. For the empirical results, a detailed description of Algorithm 1 and additional experimental results are presented in Appendix D and Appendix E, respectively. The source code used in the experiments is also provided as supplementary material.

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

## A    CONVERGENCE ANALYSIS OF DoG AND DoWG

### A.1    POSITIONING OF THIS SECTION

This section aims to show deterministic convergence bounds for both DoG Ivgi et al. (2023) and DoWG Khaled et al. (2023). Although this section does not contain our original contributions, it is included for two reasons: (i) to highlight the differences between the convergence analyses of DoG/DoWG and our analysis in Appendix C, and (ii) to provide deterministic convergence bounds for both DoG and DoWG, whereas Ivgi et al. (2023) establishes only high-probability convergence bounds.

### A.2    UPDATE RULES OF DoG AND DoWG

The update rules of DoG and DoWG are illustrated in Algorithm 2.

---

**Algorithm 2** DoG and DoWG

---

1: Initialization $\mathbf{x}^{(0)}, r^{(0)}(> 0), v^{(0)}(> 0)$
2: **if** (DoG) $\eta^{(0)} = r^{(0)}/\sqrt{v^{(0)}}$, **else if** (DoWG) $\eta^{(0)} = (r^{(0)})^2/\sqrt{v^{(0)}}$, **end**
3: **for** $t = 0, 1, \ldots, T-1$ **do**
4:     $\mathbf{x}^{(t+1)} = \mathbf{x}^{(t)} - \eta^{(t)}\nabla f(\mathbf{x}^{(t)})$
5:     $r^{(t+1)} = \max\{\|\mathbf{x}^{(t+1)} - \mathbf{x}^{(0)}\|, r^{(t)}\}$
6:     **if** (DoG) **then**
7:         $v^{(t+1)} = v^{(t)} + \|\nabla f(\mathbf{x}^{(t+1)})\|^2$
8:         $\eta^{(t+1)} = r^{(t+1)}/\sqrt{v^{(t+1)}}$
9:     **else if** (DoWG) **then**
10:        $v^{(t+1)} = v^{(t)} + (r^{(t+1)})^2\|\nabla f(\mathbf{x}^{(t+1)})\|^2$
11:        $\eta^{(t+1)} = (r^{(t+1)})^2/\sqrt{v^{(t+1)}}$
12:    **end if**
13: **end for**

---

### A.3    CONVERGENCE ANALYSIS OF DoG IVGI ET AL. (2023) AND DoWG KHALED ET AL. (2023)

First, several technical lemmas are provided.

**Lemma 4** (Lemma 3 in Ivgi et al. (2023)). *Let $s^{(0)}, s^{(1)}, \ldots, s^{(T)}$ be a positive increasing sequence. Then*

$$\max_{t \leq T} \sum_{i < t} \frac{s^{(i)}}{s^{(t)}} \geq \frac{1}{e}\Big(\frac{T}{\log_+\big(\frac{s^{(T)}}{s^{(0)}}\big)} - 1\Big),$$

*where $\log_+(x) := \log(x) + 1$.*

**Lemma 5** (Lemma 4 in Ivgi et al. (2023)). *Let $a^{(0)}, \ldots, a^{(T)}$ be a non-decreasing sequence of nonnegative numbers. Then*

$$\sum_{t=1}^{T} \frac{a^{(t)} - a^{(t-1)}}{\sqrt{a^{(t)}}} \leq 2\Big(\sqrt{a^{(T)}} - \sqrt{a^{(0)}}\Big).$$

Next, proof of Lemma 1 is given.

*Proof.* From (1), we get

$$\|\mathbf{x}^{(t+1)} - \mathbf{x}^*\|^2 \leq \|\mathbf{x}^{(t)} - \eta^{(t)}\nabla f(\mathbf{x}^{(t)}) - \mathbf{x}^*\|^2$$
$$= \|\mathbf{x}^{(t)} - \mathbf{x}^*\|^2 - 2\eta^{(t)}\langle\nabla f(\mathbf{x}^{(t)}), \mathbf{x}^{(t)} - \mathbf{x}^*\rangle + (\eta^{(t)})^2\|\nabla f(\mathbf{x}^{(t)})\|^2.$$

Rearranging this, we get

$$\langle \nabla f(\mathbf{x}^{(t)}), \mathbf{x}^{(t)} - \mathbf{x}^* \rangle \leq \frac{1}{2\eta^{(t)}} \left( \|\mathbf{x}^{(t)} - \mathbf{x}^*\|^2 - \|\mathbf{x}^{(t+1)} - \mathbf{x}^*\|^2 \right) + \frac{\eta^{(t)}}{2} \|\nabla f(\mathbf{x}^{(t)})\|^2.$$

From convexity of $f$, we get $f(\mathbf{x}^{(t)}) - f(\mathbf{x}^*) \leq \langle \nabla f(\mathbf{x}^{(t)}), \mathbf{x}^{(t)} - \mathbf{x}^* \rangle$. Integrating these inequalities results in

$$f(\mathbf{x}^{(t)}) - f(\mathbf{x}^*) \leq \frac{1}{2\eta^{(t)}} \left( (d^{(t)})^2 - (d^{(t+1)})^2 \right) + \frac{\eta^{(t)}}{2} \|\nabla f(\mathbf{x}^{(t)})\|^2,$$

where $(d^{(t)})^2 = \|\boldsymbol{x}^{(t)} - \boldsymbol{x}^*\|^2$. This results in Lemma 1. $\qquad\square$

**Lemma 6.** *Suppose that $f$ is convex, and has minimizer $\mathbf{x}^* \in \mathcal{X}$. For the iterations generated by (1), we have:*

$$[\mathbf{DoG}] \ \sum_{t=0}^{T-1} r^{(t)} \Big( f(\mathbf{x}^{(t+1)}) - f(\mathbf{x}^*) \Big) \leq \underbrace{\sum_{t=0}^{T-1} \frac{r^{(t)}}{2\eta^{(t)}} \left( (d^{(t)})^2 - (d^{(t+1)})^2 \right)}_{(A_1)} + \underbrace{\sum_{t=0}^{T-1} \frac{r^{(t)}\eta^{(t)}}{2} \left\| \nabla f(\mathbf{x}^{(t)}) \right\|^2}_{(B_1)},$$

$$[\mathbf{DoWG}] \ \sum_{t=0}^{T-1} (r^{(t)})^2 \Big( f(\mathbf{x}^{(t+1)}) - f(\mathbf{x}^*) \Big) \leq \underbrace{\sum_{t=0}^{T-1} \frac{(r^{(t)})^2}{2\eta^{(t)}} \left( (d^{(t)})^2 - (d^{(t+1)})^2 \right)}_{(A_2)} + \underbrace{\sum_{t=0}^{T-1} \frac{(r^{(t)})^2\eta^{(t)}}{2} \left\| \nabla f(\mathbf{x}^{(t)}) \right\|^2}_{(B_2)}.$$

*Proof.* The weighted sum of Lemma 1 results in the statement. $\qquad\square$

**Lemma 7.** *Suppose that $f$ is a convex function with a minimizer $\boldsymbol{x}^*$. For the iterations generated by Algorithm 2, we have:*

$$[\mathbf{DoG}] \qquad \sum_{t=0}^{T-1} r^{(t)} \Big( f(\mathbf{x}^{(t)}) - f(\mathbf{x}^*) \Big) \leq 2r^{(T)} \Big[ \overline{d}^{(T)} + r^{(T)} \Big] \sqrt{v^{(T-1)}},$$

$$[\mathbf{DoWG}] \quad \sum_{t=0}^{T-1} (r^{(t)})^2 \Big( f(\mathbf{x}^{(t)}) - f(\mathbf{x}^*) \Big) \leq 2r^{(T)} \Big[ \overline{d}^{(T)} + r^{(T)} \Big] \sqrt{v^{(T-1)}},$$

*where $\overline{d}^{(T)} = \max_{t \leq T} d^{(t)}$.*

*Proof.* For RHS term for DoG in Lemma 6, we get

$$
\begin{aligned}
2A_1 &= \sum_{t=0}^{T-1} \frac{r^{(t)}}{\eta^{(t)}} \big( (d^{(t)})^2 - (d^{(t+1)})^2 \big) \\
&= \sum_{t=0}^{T-1} \sqrt{v^{(t)}} \big( (d^{(t)})^2 - (d^{(t+1)})^2 \big) \\
&= \big( d^{(0)} \big)^2 \sqrt{v^{(0)}} - \big( d^{(T)} \big)^2 \sqrt{v^{(T-1)}} + \sum_{t=1}^{T-1} \big( d^{(t)} \big)^2 \big( \sqrt{v^{(t)}} - \sqrt{v^{(t-1)}} \big) \\
&\leq \big( \overline{d}^{(T)} \big)^2 \sqrt{v^{(0)}} - \big( d^{(T)} \big)^2 \sqrt{v^{(T-1)}} + \big( \overline{d}^{(T)} \big)^2 \sum_{t=1}^{T-1} \big( \sqrt{v^{(t)}} - \sqrt{v^{(t-1)}} \big) \\
&\leq \sqrt{v^{(T-1)}} \Big( \big( \overline{d}^{(T)} \big)^2 - \big( d^{(T)} \big)^2 \Big) \\
&\leq 4r^{(T)} \overline{d}^{(T)} \sqrt{v^{(T-1)}},
\end{aligned}
$$

where $(\overline{d}^{(t)})^2 - (d^{(t)})^2 = (\overline{d}^{(t)} + d^{(t)})(\overline{d}^{(t)} - d^{(t)}) \leq \|\overline{d}^{(t)} - d^{(t)}\|(\overline{d}^{(t)} + d^{(t)}) \leq 4r^{(t)}\overline{d}^{(t)}$ is used in the last line.

For another RHS term for DoG in Lemma 6, we get

$$2B_1 = \sum_{t=0}^{T-1} r^{(t)} \eta^{(t)} \left\| \nabla f(\mathbf{x}^{(t)}) \right\|^2$$

$$= (r^{(0)})^2 \sqrt{v^{(0)}} + \sum_{t=1}^{T-1} \frac{(r^{(t)})^2}{\sqrt{v^{(t)}}} \left\| \nabla f(\mathbf{x}^{(t)}) \right\|^2$$

$$\leq (r^{(T)})^2 \sqrt{v^{(0)}} + (r^{(T)})^2 \sum_{t=1}^{T-1} \frac{1}{\sqrt{v^{(t)}}} \left\| \nabla f(\mathbf{x}^{(t)}) \right\|^2$$

$$= (r^{(T)})^2 \sqrt{v^{(0)}} + (r^{(T)})^2 \sum_{t=1}^{T-1} \frac{v^{(t)} - v^{(t-1)}}{\sqrt{v^{(t)}}}$$

$$\leq (r^{(T)})^2 \sqrt{v^{(0)}} + (r^{(T)})^2 \left[ \sqrt{v^{(T-1)}} - \sqrt{v^{(0)}} \right]$$

$$\leq 2(r^{(T)})^2 \sqrt{v^{(T-1)}}.$$

Integrating these inequalities and Lemma 6 results in

$$\sum_{t=0}^{T-1} r^{(t)} \left( f(\mathbf{x}^{(t)}) - f(\mathbf{x}^*) \right) \leq 2 r^{(T)} \left[ \overline{d}^{(T)} + r^{(T)} \right] \sqrt{v^{(T-1)}}.$$

For RHS term for DoWG in Lemma 6, we get

$$2A_2 = \sum_{t=0}^{T-1} \frac{(r^{(t)})^2}{\eta^{(t)}} \left( (d^{(t)})^2 - (d^{(t+1)})^2 \right)$$

$$= \sum_{t=0}^{T-1} \sqrt{v^{(t)}} \left( (d^{(t)})^2 - (d^{(t+1)})^2 \right)$$

$$= \left( d^{(0)} \right)^2 \sqrt{v^{(0)}} - \left( d^{(T)} \right)^2 \sqrt{v^{(T-1)}} + \sum_{t=1}^{T-1} \left( d^{(t)} \right)^2 \left( \sqrt{v^{(t)}} - \sqrt{v^{(t-1)}} \right)$$

$$\leq \left( \overline{d}^{(T)} \right)^2 \sqrt{v^{(0)}} - \left( d^{(T)} \right)^2 \sqrt{v^{(T-1)}} + \left( \overline{d}^{(T)} \right)^2 \sum_{t=1}^{T-1} \left( \sqrt{v^{(t)}} - \sqrt{v^{(t-1)}} \right)$$

$$\leq \sqrt{v^{(T-1)}} \left( \left( \overline{d}^{(T)} \right)^2 - \left( d^{(T)} \right)^2 \right)$$

$$\leq 4 r^{(T)} \overline{d}^{(T)} \sqrt{v^{(T-1)}},$$

where $(\overline{d}^{(t)})^2 - (d^{(t)})^2 = (\overline{d}^{(t)} + d^{(t)})(\overline{d}^{(t)} - d^{(t)}) \leq \| \overline{d}^{(t)} - d^{(t)} \| (\overline{d}^{(t)} + d^{(t)}) \leq 4 r^{(t)} \overline{d}^{(t)}$ is used in the last line.

For another RHS term for DoWG in Lemma 6, we get

$$2B_2 = \sum_{t=0}^{T-1} (r^{(t)})^2 \eta^{(t)} \left\| \nabla f(\mathbf{x}^{(t)}) \right\|^2$$

$$= (r^{(0)})^2 \sqrt{v^{(0)}} + \sum_{t=1}^{T-1} \frac{(r^{(t)})^4}{\sqrt{v^{(t)}}} \left\| \nabla f(\mathbf{x}^{(t)}) \right\|^2$$

$$\leq (r^{(T)})^2 \sqrt{v^{(0)}} + (r^{(T)})^2 \sum_{t=1}^{T-1} \frac{(r^{(t)})^2}{\sqrt{v^{(t)}}} \left\| \nabla f(\mathbf{x}^{(t)}) \right\|^2$$

$$= (r^{(T)})^2 \sqrt{v^{(0)}} + (r^{(T)})^2 \sum_{t=1}^{T-1} \frac{v^{(t)} - v^{(t-1)}}{\sqrt{v^{(t)}}}$$

$$\leq (r^{(T)})^2 \sqrt{v^{(0)}} + (r^{(T)})^2 \left[ \sqrt{v^{(T-1)}} - \sqrt{v^{(0)}} \right]$$

$$\leq 2(r^{(T)})^2 \sqrt{v^{(T-1)}}.$$

Integrating these inequalities and Lemma 6 results in

$$\sum_{t=0}^{T-1} (r^{(t)})^2 \Big( f(\mathbf{x}^{(t)}) - f(\mathbf{x}^*) \Big) \leq 2r^{(T)} \left[ \overline{d}^{(T)} + r^{(T)} \right] \sqrt{v^{(T-1)}}.$$

$\square$

Finally, proof of Theorem 1 is shown.

*Proof.* Under the assumption of $G$-Lipschitz function $f$, Lemma 7 for DoG is reformulated as

$$\sum_{t=0}^{T-1} r^{(t)} \Big( f(\mathbf{x}^{(t)}) - f(\mathbf{x}^*) \Big) \leq 2r^{(T)} [\overline{d}^{(T)} + r^{(T)}] \sqrt{\sum_{t=0}^{T-1} \left\| \nabla f(\mathbf{x}^{(t)}) \right\|^2}$$

$$\leq 2r^{(T)} [\overline{d}^{(T)} + r^{(T)}] G \sqrt{T}.$$

By normalizing this, we get

$$\frac{1}{\sum_{t=0}^{T-1} r^{(t)}} \sum_{t=0}^{T-1} r^{(t)} \Big( f(\mathbf{x}^{(t)}) - f(\mathbf{x}^*) \Big) \leq \frac{2[\overline{d}^{(T)} + r^{(T)}] G \sqrt{T}}{\sum_{t=0}^{T-1} \frac{r^{(t)}}{r^{(T)}}}.$$

Using Lemma 4, we get

$$\sum_{t=0}^{T-1} \frac{r^{(t)}}{r^{(T)}} \geq \frac{1}{e} \left( \frac{T}{\log_+ \frac{r^{(T)}}{r^{(0)}}} - 1 \right) \geq \frac{1}{e} \left( \frac{T}{\log_+ \frac{D_0}{r^{(0)}}} - 1 \right).$$

We now have two cases:

(i) If $T \geq 2 \log_+ \frac{D}{r^{(0)}}$, i.e., $\frac{T}{\log_+ \frac{D}{r^{(0)}}} - 1 \geq \frac{T}{2 \log_+ \frac{D}{r^{(0)}}}$, we get

$$f(\widehat{\mathbf{x}}^{(t)}) - f(\mathbf{x}^*) \leq \frac{1}{\sum_{t=0}^{T-1} r^{(t)}} \sum_{t=0}^{T-1} r^{(t)} \Big( f(\mathbf{x}^{(t)}) - f(\mathbf{x}^*) \Big)$$

$$\leq \frac{8[\overline{d}^{(T)} + r^{(T)}] G}{\sqrt{T}} \log \frac{r^{(T)}}{r^{(0)}}$$

$$\leq \frac{16 D_0 G}{\sqrt{T}} \log \frac{D_0}{r^{(0)}}$$

$$= \widetilde{\mathcal{O}} \Big( \frac{D_0 G}{\sqrt{T}} \Big),$$

where we used $r^{(T)} \le D_0$ and $\overline{d}^{(T)} \le D_0$ because the diameter of $\mathcal{X}$ is bounded by $D_0$.

(ii) If $T < 2 \log_+ \frac{D}{r^{(0)}}$, i.e., $1 < \frac{2 \log_+ \frac{D}{r^{(0)}}}{T}$, we get following by using Cauchy-Schwarz.

$$
\begin{aligned}
f(\widehat{\mathbf{x}}^{(t)}) - f(\mathbf{x}^*) &\le \langle \nabla f(\widehat{\mathbf{x}}^{(t)}), \widehat{\mathbf{x}}^{(t)} - \mathbf{x}^* \rangle \\
&\le \|\nabla f(\widehat{\mathbf{x}}^{(t)})\| \|\widehat{\mathbf{x}}^{(t)} - \mathbf{x}^*\| \\
&\le GD \\
&\le \frac{2GD \log_+ \frac{D}{r^{(0)}}}{T} \\
&\le \frac{2GD \log_+ \frac{D}{r^{(0)}}}{\sqrt{T}} \\
&= \widetilde{\mathcal{O}}\Big(\frac{D_0 G}{\sqrt{T}}\Big).
\end{aligned}
$$

Similarly, Lemma 7 for DoWG can be reformulated under the assumption of $G$-Lipschitz function $f$:

$$
\begin{aligned}
\sum_{t=0}^{T-1} (r^{(t)})^2 \Big( f(\mathbf{x}^{(t)}) - f(\mathbf{x}^*) \Big) &\le 2r^{(T)} [\overline{d}^{(T)} + r^{(T)}] \sqrt{\sum_{t=0}^{T-1} (r^{(t)})^2 \left\| \nabla f(\mathbf{x}^{(t)}) \right\|^2} \\
&\le 2r^{(T)} [\overline{d}^{(T)} + r^{(T)}] \sqrt{(r^{(T)})^2 \sum_{t=0}^{T-1} \left\| \nabla f(\mathbf{x}^{(t)}) \right\|^2} \\
&\le 2(r^{(T)})^2 [\overline{d}^{(T)} + r^{(T)}] G \sqrt{T}.
\end{aligned}
$$

By normalizing this, we get

$$
\frac{1}{\sum_{t=0}^{T-1} (r^{(t)})^2} \sum_{t=0}^{T-1} (r^{(t)})^2 \Big( f(\mathbf{x}^{(t)}) - f(\mathbf{x}^*) \Big) \le \frac{2[\overline{d}^{(T)} + r^{(T)}] G \sqrt{T}}{\sum_{t=0}^{T-1} \Big( \frac{r^{(t)}}{r^{(T)}} \Big)^2}.
$$

Using Lemma 4, we get

$$
\sum_{t=0}^{T-1} \Big( \frac{r^{(t)}}{r^{(T)}} \Big)^2 \ge \frac{1}{e} \left( \frac{T}{\log_+ \Big( \frac{r^{(T)}}{r^{(0)}} \Big)^2} - 1 \right) \ge \frac{1}{e} \left( \frac{T}{\log_+ \Big( \frac{D_0}{r^{(0)}} \Big)^2} - 1 \right).
$$

We now have two cases:

(i) If $T \ge 2 \log_+ \Big( \frac{D}{r^{(0)}} \Big)^2$, i.e., $\frac{T}{\log_+ \big( \frac{D}{r^{(0)}} \big)^2} - 1 \ge \frac{T}{2 \log_+ \big( \frac{D}{r^{(0)}} \big)^2}$, we get

$$
\begin{aligned}
f(\widehat{\mathbf{x}}^{(t)}) - f(\mathbf{x}^*) &\le \frac{1}{\sum_{t=0}^{T-1} (r^{(t)})^2} \sum_{t=0}^{T-1} (r^{(t)})^2 \Big( f(\mathbf{x}^{(t)}) - f(\mathbf{x}^*) \Big) \\
&\le \frac{8[\overline{d}^{(T)} + r^{(T)}] G}{\sqrt{T}} \log \frac{r^{(T)}}{r^{(0)}} \\
&\le \frac{16 D_0 G}{\sqrt{T}} \log \frac{D_0}{r^{(0)}} \\
&= \widetilde{\mathcal{O}}\Big(\frac{D_0 G}{\sqrt{T}}\Big),
\end{aligned}
$$

where we used $r^{(T)} \le D_0$ and $\overline{d}^{(T)} \le D_0$ because the diameter of $\mathcal{X}$ is bounded by $D_0$.

(ii) If $T < 2\log_+\left(\frac{D}{r^{(0)}}\right)^2$, i.e., $1 < \frac{2\log_+\left(\frac{D}{r^{(0)}}\right)^2}{T}$, we get following by using Cauchy-Schwarz.

$$
\begin{aligned}
f(\widehat{\mathbf{x}}^{(t)}) - f(\mathbf{x}^*) &\leq \langle \nabla f(\widehat{\mathbf{x}}^{(t)}), \widehat{\mathbf{x}}^{(t)} - \mathbf{x}^* \rangle \\
&\leq \|\nabla f(\widehat{\mathbf{x}}^{(t)})\|\|\widehat{\mathbf{x}}^{(t)} - \mathbf{x}^*\| \\
&\leq GD \\
&\leq \frac{2GD\log_+\left(\frac{D}{r^{(0)}}\right)^2}{T} \\
&\leq \frac{4GD\log_+\frac{D}{r^{(0)}}}{\sqrt{T}} \\
&= \tilde{\mathcal{O}}\left(\frac{D_0 G}{\sqrt{T}}\right).
\end{aligned}
$$

$\square$

## B  AN INTERPRETATION OF APPROXIMATED SOLUTION OF LOCAL UPDATE IN FEDPROX

Local update rule in (3) is rewritten as

$$\mathbf{x}_i^{(t+1)} = \arg\min_{\mathbf{y}\in\chi}\Big(f_i(\mathbf{y}) + \frac{\mu}{2}\left\|\mathbf{y} - \mathbf{x}^{(t)}\right\|^2\Big).$$

Approximating local loss function around current model $\mathbf{x}_i^{(t)}$ as $f_i(\mathbf{y}) \approx f_i(\mathbf{x}_i^{(t)}) + \langle\nabla f_i(\mathbf{x}_i^{(t)}), \mathbf{y} - \mathbf{x}_i^{(t)}\rangle + \frac{1}{2\eta'}\|\mathbf{y} - \mathbf{x}_i^{(t)}\|^2$, an approximated solution is obtained as

$$\nabla f_i(\mathbf{x}_i^{(t)}) + \frac{1}{\eta'}\big(\mathbf{y} - \mathbf{x}_i^{(t)}\big) + \mu\big(\mathbf{y} - \mathbf{x}^{(t)}\big) = 0,$$

namely, we get

$$\mathbf{y} = \mathbf{x}_i^{(t)} - \frac{\eta'}{1 + \eta'\mu}\big(\nabla f_i(\mathbf{x}_i^{(t)}) + \mu\big(\mathbf{x}_i^{(t)} - \mathbf{x}^{(t)}\big)\big).$$

Replacing $\eta' = \frac{\eta}{1+\eta\mu}$, and $\mathbf{y} = \mathbf{x}_i^{(t+1)}$ results in

$$\mathbf{x}_i^{(t+1)} = \mathbf{x}_i^{(t)} - \eta\big(\nabla f_i(\mathbf{x}_i^{(t)}) + \mu\big(\mathbf{x}_i^{(t)} - \mathbf{x}^{(t)}\big)\big).$$

## C  CONVERGENCE ANALYSIS FOR PROPOSED ALGORITHMS

Convergence analysis for the proposed algorithms, FedProxLoD and FedProxWLoD, introduced in Section 4, is provided.

### C.1  PROOFS FOR THEOREM 2

**Why we begin with FedProx.**    Our primary interest lies in parameter-free optimization. Among many approaches, DoG and DoWG have performed well empirically even with non-convex DNNs; our main idea is to leverage these successes to Federated Learning (FL). Within this context, we examine which fundamental FL algorithms (e.g., FedAvg, FedProx) yield an inequality analogous to Lemma 1–the key lemma underlying DoG/DoWG. We find that this holds only for FedProx, which leads to Lemma 2. Upon reconsideration, we conclude that Lemma 1 mirrors the descent inequality of the classical proximal point method, which FedProx implicitly includes.

**Proof sketch.**    Focusing on the local parameter update in FedProxLoD and FedProxWLoD, we derive Lemma 2, which shares structural similarity with Lemma 1. The latter is used for adaptively determining the learning rate $\eta^{(r)}$ in (2), as well as for the convergence analysis in DoG and DoWG. Leveraging this similarity, we determine the proximal weight via (6). Based on Lemma 2 and the proximal weight expression in (6), we derive Lemmas 8 and 3. Furthermore, under additional Assumptions 4–5, the loss difference appearing in Lemma 2 can be bounded as shown in Lemma 11. By using these lemmas, we get Theorem 2.

First, proof of Lemma 2 is given.

*Proof.* Analogous to (4), the local parameter update in FedProxLoD and FedProxWLoD (Line 6 of Algorithm 1) yields

$$\nabla f_i(\mathbf{x}_i^{(t+1)}) + \mu^{(t)}(\mathbf{x}_i^{(t+1)} - \mathbf{x}^{(t)}) \in -\partial\Pi_{\mathcal{X}}(\mathbf{x}_i^{(t+1)}).$$

From (5), we have that $\langle\partial\Pi_{\mathcal{X}}(\mathbf{a}), \mathbf{a} - \mathbf{b}\rangle = 0$ for any two points $\mathbf{a}, \mathbf{b} \in \mathbb{R}^m$. Using this property, we obtain

$$\left\langle\nabla f_i(\mathbf{x}_i^{(t+1)}), \mathbf{x}_i^{(t+1)} - \mathbf{x}^*\right\rangle \leq -\mu^{(t)}\left\langle\mathbf{x}_i^{(t+1)} - \mathbf{x}^{(t)}, \mathbf{x}_i^{(t+1)} - \mathbf{x}^*\right\rangle$$

$$= \frac{\mu^{(t)}}{2}\left(\|\mathbf{x}^{(t)} - \mathbf{x}^*\|^2 - \|\mathbf{x}_i^{(t+1)} - \mathbf{x}^{(t)}\|^2 - \|\mathbf{x}_i^{(t+1)} - \mathbf{x}^*\|^2\right).$$

From convexity of $f_i$, $f_i(\mathbf{x}_i^{(t+1)}) - f_i(\mathbf{x}^*) \leq \left\langle\nabla f_i(\mathbf{x}_i^{(t+1)}), \mathbf{x}_i^{(t+1)} - \mathbf{x}^*\right\rangle$; thus, we obtain

$$f_i(\mathbf{x}_i^{(t+1)}) - f_i(\mathbf{x}^*) \leq \frac{\mu^{(t)}}{2}\left(\|\mathbf{x}^{(t)} - \mathbf{x}^*\|^2 - \|\mathbf{x}_i^{(t+1)} - \mathbf{x}^{(t)}\|^2 - \|\mathbf{x}_i^{(t+1)} - \mathbf{x}^*\|^2\right). \quad (7)$$

Adding $f_i(\mathbf{x}^{(t+1)}) - f_i(\mathbf{x}_i^{(t+1)})$ on both sides results in

$$f_i(\mathbf{x}^{(t+1)}) - f_i(\mathbf{x}^*)$$

$$\leq \frac{\mu^{(t)}}{2}\left(\|\mathbf{x}^{(t)} - \mathbf{x}^*\|^2 - \|\mathbf{x}_i^{(t+1)} - \mathbf{x}^*\|^2\right) + f_i(\mathbf{x}^{(t+1)}) - f_i(\mathbf{x}_i^{(t+1)}) - \frac{\mu^{(t)}}{2}\|\mathbf{x}_i^{(t+1)} - \mathbf{x}^{(t)}\|^2.$$

Averaging over $n$ clients, we get

$$f(\mathbf{x}^{(t+1)}) - f(\mathbf{x}^*)$$

$$\leq \frac{\mu^{(t)}}{2}\left(\|\mathbf{x}^{(t)} - \mathbf{x}^*\|^2 - \frac{1}{n}\sum_{i=1}^n\|\mathbf{x}_i^{(t+1)} - \mathbf{x}^*\|^2\right) + f(\mathbf{x}^{(t+1)}) - \frac{1}{n}\sum_{i=1}^n f_i(\mathbf{x}_i^{(t+1)}) - \frac{\mu^{(t)}}{2n}\sum_{i=1}^n\|\mathbf{x}_i^{(t+1)} - \mathbf{x}^{(t)}\|^2$$

$$\leq \frac{\mu^{(t)}}{2}\Big(\underbrace{\|\mathbf{x}^{(t)} - \mathbf{x}^*\|^2}_{d^{(t)}} - \underbrace{\|\mathbf{x}^{(t+1)} - \mathbf{x}^*\|^2}_{d^{(t+1)}}\Big) + \underbrace{\max\left\{f(\mathbf{x}^{(t+1)}) - \frac{1}{n}\sum_{i=1}^n f_i(\mathbf{x}_i^{(t+1)}) - \frac{\mu^{(t)}}{2n}\sum_{i=1}^n\|\mathbf{x}_i^{(t+1)} - \mathbf{x}^{(t)}\|^2, 0\right\}}_{\Delta^{(t+1)}}.$$

$\square$

**Lemma 8.** *Suppose that Assumptions 1-3, and 6 hold. We have:* $\|\mathbf{x}^{(t+1)} - \mathbf{x}^*\| \leq \|\mathbf{x}^{(t)} - \mathbf{x}^*\|$.

*Proof.* We rewrite (7),

$$f_i(\mathbf{x}_i^{(t+1)}) - f_i(\mathbf{x}^*) \leq \frac{\mu^{(t)}}{2}\Big(\|\mathbf{x}^{(t)} - \mathbf{x}^*\|^2 - \|\mathbf{x}_i^{(t+1)} - \mathbf{x}^{(t)}\|^2 - \|\mathbf{x}_i^{(t+1)} - \mathbf{x}^*\|^2\Big).$$

Under Assumption 6, $\mathbf{x}^*$ is optimal solution of $f_i$. Thus, LHS holds $f_i(\mathbf{x}_i^{(t+1)}) - f_i(\mathbf{x}^*) \geq 0$. Thus, we get

$$\|\mathbf{x}_i^{(t+1)} - \mathbf{x}^*\|^2 \leq \|\mathbf{x}^{(t)} - \mathbf{x}^*\|^2.$$

$\square$

Next, proof of Lemma 3 is given.

*Proof.* First, we rewrite Lemma 2:

$$f(\mathbf{x}^{(t+1)}) - f(\mathbf{x}^*) \leq \frac{\mu^{(t)}}{2}\Big(d^{(t)} - d^{(t+1)}\Big) + \frac{1}{\mu^{(t)}}\Delta^{(t+1)},$$

where all terms on RHS are non-negative, specifically, $\Delta^{(t+1)} \geq 0$ and $\mu^{(t)} \geq 0$, and $d^{(t)} - d^{(t+1)} \geq 0$ under Assumption 6 by Lemma 8. To facilitate the application of convergence analysis techniques from DoG and DoWG, we aim to reformulate the second term on the RHS from $\frac{1}{\mu^{(t)}}\Delta^{(t+1)}$ to $\frac{1}{\mu^{(t+1)}}\Delta^{(t+1)}$. For this aim, $\min\{\frac{\mu^{(t)}}{\mu^{(t+1)}}, 1\} \in (0, 1]$ is multiplied to both side, yielding

$$\min\left\{\frac{\mu^{(t)}}{\mu^{(t+1)}}, 1\right\}\Big(f(\mathbf{x}^{(t+1)}) - f(\mathbf{x}^*)\Big)$$

$$\leq \min\left\{\frac{\mu^{(t)}}{\mu^{(t+1)}}, 1\right\}\frac{\mu^{(t)}}{2}\Big(d^{(t)} - d^{(t+1)}\Big) + \min\left\{\frac{\mu^{(t)}}{\mu^{(t+1)}}, 1\right\}\frac{1}{\mu^{(t)}}\Delta^{(t+1)}$$

$$\leq \frac{\mu^{(t)}}{2}\Big(d^{(t)} - d^{(t+1)}\Big) + \frac{1}{\mu^{(t+1)}}\Delta^{(t+1)},$$

where we used $\min\{\frac{\mu^{(t)}}{\mu^{(t+1)}}, 1\}\frac{1}{\mu^{(t)}} \leq \frac{1}{\mu^{(t+1)}}$ in the third line, because:

(i) when $\frac{\mu^{(t)}}{\mu^{(t+1)}} \leq 1$, we have

$$\min\{\frac{\mu^{(t)}}{\mu^{(t+1)}}, 1\} = \frac{\mu^{(t)}}{\mu^{(t+1)}}, \quad \text{and} \quad \frac{\mu^{(t)}}{\mu^{(t+1)}} \cdot \frac{1}{\mu^{(t)}} = \frac{1}{\mu^{(t+1)}},$$

(ii) when $\frac{\mu^{(t)}}{\mu^{(t+1)}} > 1$ (i.e., $\mu^{(t)} > \mu^{(t+1)}$), we have

$$\min\{\frac{\mu^{(t)}}{\mu^{(t+1)}}, 1\} = 1 \quad \text{and} \quad 1 \cdot \frac{1}{\mu^{(t)}} = \frac{1}{\mu^{(t)}} \leq \frac{1}{\mu^{(t+1)}}.$$

Similar to Lemma 6 for DoG and DoWG, two patterns of weighted sum over $T$ iterations result in

**[FedProxLoD]**

$$\sum_{k=0}^{T-1} \min\left\{\frac{\mu^{(k)}}{\mu^{(k+1)}}, 1\right\} r^{(k)}\Big(f(\mathbf{x}^{(k+1)}) - f(\mathbf{x}^*)\Big) \leq \underbrace{\sum_{k=0}^{T-1} \frac{r^{(k)}\mu^{(k)}}{2}\Big(d^{(k)} - d^{(k+1)}\Big)}_{(A_1')} + \underbrace{\sum_{k=0}^{T-1} \frac{r^{(k)}}{\mu^{(k+1)}}\Delta^{(k+1)}}_{(B_1')},$$

**[FedProxWLoD]**

$$\sum_{k=0}^{T-1} \min\left\{\frac{\mu^{(k)}}{\mu^{(k+1)}}, 1\right\} (r^{(k)})^2\Big(f(\mathbf{x}^{(k+1)}) - f(\mathbf{x}^*)\Big) \leq \underbrace{\sum_{k=0}^{T-1} \frac{(r^{(k)})^2\mu^{(k)}}{2}\Big(d^{(k)} - d^{(k+1)}\Big)}_{(A_2')} + \underbrace{\sum_{k=0}^{T-1} \frac{(r^{(k)})^2}{\mu^{(k+1)}}\Delta^{(k+1)}}_{(B_2')}.$$

$\square$

**Lemma 9.** *Suppose that Assumptions 1-6 hold. For the iterations generated by Line 6 of Algorithm 1, we have:*

[**FedProxLoD**]

$$\sum_{k=0}^{T-1} \min\left\{\frac{\mu^{(k)}}{\mu^{(k+1)}}, 1\right\} r^{(k)} \left(f(\mathbf{x}^{(k+1)}) - f(\mathbf{x}^*)\right) \leq 2r^{(T)}\left(\overline{d}^{(T)} + r^{(T)}\right)\sqrt{u^{(T)}},$$

[**FedProxWLoD**]

$$\sum_{k=0}^{T-1} \min\left\{\frac{\mu^{(k)}}{\mu^{(k+1)}}, 1\right\} (r^{(k)})^2 \left(f(\mathbf{x}^{(k+1)}) - f(\mathbf{x}^*)\right) \leq 2r^{(T)}\left(\overline{d}^{(T)} + r^{(T)}\right)\sqrt{u^{(T)}}.$$

*Proof.* For FedProxLoD, proximal weight is determined by $\mu^{(k)} = \sqrt{u^{(k)}}/r^{(k)}$ in (6). By using $\overline{d}^{(t)} = \max_{k \leq t} d^{(k)}$, RHS term $(A_1')$ in Lemma 3 can be bounded as

$$
\begin{aligned}
A_1' &= \sum_{k=0}^{T-1} \frac{r^{(k)}\mu^{(k)}}{2}\left((d^{(k)})^2 - (d^{(k+1)})^2\right) \\
&= \sum_{k=0}^{T-1} \frac{\sqrt{u^{(k)}}}{2}\left((d^{(k)})^2 - (d^{(k+1)})^2\right) \\
&= \frac{1}{2}\left\{\left(d^{(0)}\right)^2\sqrt{u^{(0)}} - \left(d^{(T)}\right)^2\sqrt{u^{(T-1)}} + \sum_{k=1}^{T-1}\left(d^{(k)}\right)^2\left(\sqrt{u^{(k)}} - \sqrt{u^{(k-1)}}\right)\right\} \\
&\leq \frac{1}{2}\left\{\left(d^{(0)}\right)^2\sqrt{u^{(0)}} - \left(d^{(T)}\right)^2\sqrt{u^{(T-1)}} + \left(\overline{d}^{(T-1)}\right)^2\left(\sqrt{u^{(T-1)}} - \sqrt{u^{(0)}}\right)\right\} \\
&\leq \frac{1}{2}\left\{\left(\overline{d}^{(T)}\right)^2 - \left(d^{(T)}\right)^2\right\}\sqrt{u^{(T-1)}} \\
&= 2r^{(T)}\overline{d}^{(T)}\sqrt{u^{(T-1)}} \\
&\leq 2r^{(T)}\overline{d}^{(T)}\sqrt{u^{(T)}},
\end{aligned}
$$

where techniques from Lemma 7 in Appendix A are employed.

Using $u^{(t+1)} = u^{(t)} + \Delta^{(t+1)}$, RHS term $(B_1')$ in Lemma 3 can be bounded as

$$
\begin{aligned}
B_1' &= \sum_{k=0}^{T-1} \frac{r^{(k)}}{\mu^{(k+1)}}\Delta^{(k+1)} \\
&\leq \sum_{k=0}^{T-1} \frac{r^{(k+1)}}{\mu^{(k+1)}}\Delta^{(k+1)} \\
&= \sum_{k=0}^{T-1} \frac{(r^{(k+1)})^2}{\sqrt{u^{(k+1)}}}\Delta^{(k+1)} \\
&\leq (r^{(T)})^2 \sum_{k=0}^{T-1} \frac{1}{\sqrt{u^{(k+1)}}}\Delta^{(k+1)} \\
&= (r^{(T)})^2 \sum_{k=0}^{T-1} \frac{u^{(k+1)} - u^{(k)}}{\sqrt{u^{(k+1)}}} \\
&\leq 2(r^{(T)})^2\left(\sqrt{u^{(T)}} - \sqrt{u^{(0)}}\right) \\
&\leq 2(r^{(T)})^2\sqrt{u^{(T)}},
\end{aligned}
$$

where techniques from Lemma 5 in Appendix A are employed.

Using above inequalities, Lemma 3 for FedProxLoD is reformulated as

$$\sum_{k=0}^{T-1} \min\left\{\frac{\mu^{(k)}}{\mu^{(k+1)}}, 1\right\} r^{(k)} \left(f(\mathbf{x}^{(k+1)}) - f(\mathbf{x}^*)\right) \le 2r^{(T)}\left(\overline{d}^{(T)} + r^{(T)}\right)\sqrt{u^{(T)}}.$$

On the contrary, for FedProxWLoD, proximal weight is determined by $\mu^{(k)} = \sqrt{u^{(k)}}/(r^{(k)})^2$ in (6). By using $\overline{d}^{(t)} = \max_{k \le t} d^{(k)}$, RHS term $(A_2')$ in Lemma 3 can be bounded as

$$
\begin{aligned}
A_2' &= \sum_{k=0}^{T-1} \frac{(r^{(k)})^2 \mu^{(k)}}{2}\left((d^{(k)})^2 - (d^{(k+1)})^2\right) \\
&= \sum_{k=0}^{T-1} \frac{\sqrt{u^{(k)}}}{2}\left((d^{(k)})^2 - (d^{(k+1)})^2\right) \\
&= \frac{1}{2}\left\{\left(d^{(0)}\right)^2 \sqrt{u^{(0)}} - \left(d^{(T)}\right)^2 \sqrt{u^{(T-1)}} + \sum_{k=1}^{T-1}\left(d^{(k)}\right)^2\left(\sqrt{u^{(k)}} - \sqrt{u^{(k-1)}}\right)\right\} \\
&\le \frac{1}{2}\left\{\left(d^{(0)}\right)^2 \sqrt{u^{(0)}} - \left(d^{(T)}\right)^2 \sqrt{u^{(T-1)}} + \left(\overline{d}^{(T-1)}\right)^2\left(\sqrt{u^{(T-1)}} - \sqrt{u^{(0)}}\right)\right\} \\
&\le \frac{1}{2}\left\{\left(\overline{d}^{(T)}\right)^2 - \left(d^{(T)}\right)^2\right\}\sqrt{u^{(T-1)}} \\
&= 2r^{(T)}\overline{d}^{(T)}\sqrt{u^{(T-1)}} \\
&\le 2r^{(T)}\overline{d}^{(T)}\sqrt{u^{(T)}},
\end{aligned}
$$

where techniques from Lemma 7 in Appendix A are employed.

Using $u^{(t+1)} = u^{(t)} + (r^{(t+1)})^2 \Delta^{(t+1)}$, RHS term $(B_2')$ in Lemma 3 can be bounded as

$$
\begin{aligned}
B_2' &= \sum_{k=0}^{T-1} \frac{(r^{(k)})^2}{\mu^{(k+1)}} \Delta^{(k+1)} \\
&= (r^{(0)})^2 \sqrt{u^{(0)}} + \sum_{k=1}^{T-1} \frac{(r^{(k+1)})^4}{\sqrt{u^{(k+1)}}} \Delta^{(k+1)} \\
&\le (r^{(T)})^2 \sqrt{u^{(0)}} + (r^{(T)})^2 \sum_{k=1}^{T-1} \frac{(r^{(k+1)})^2}{\sqrt{u^{(k+1)}}} \Delta^{(k+1)} \\
&= (r^{(T)})^2 \sqrt{u^{(0)}} + (r^{(T)})^2 \sum_{k=1}^{T-1} \frac{u^{(k+1)} - u^{(k)}}{\sqrt{u^{(k+1)}}} \\
&\le (r^{(T)})^2 \sqrt{u^{(0)}} + 2(r^{(T)})^2(\sqrt{u^{(T)}} - \sqrt{u^{(0)}}) \\
&\le 2(r^{(T)})^2 \sqrt{u^{(T)}},
\end{aligned}
$$

where techniques from Lemma 7 in Appendix A are employed.

Using above inequalities, Lemma 3 for FedProxWLoD is reformulated as

$$\sum_{k=0}^{T-1} \min\left\{\frac{\mu^{(k)}}{\mu^{(k+1)}}, 1\right\} (r^{(k)})^2 \left(f(\mathbf{x}^{(k+1)}) - f(\mathbf{x}^*)\right) \le 2r^{(T)}\left(\overline{d}^{(T)} + r^{(T)}\right)\sqrt{u^{(T)}}.$$

□

**Lemma 10.** *Suppose that Assumptions 1, 2, 5 hold. For the iterations generated by Line 6 of Algorithm 1, we have:*

$$\|\mathbf{x}^{(t+1)} - \mathbf{x}_i^{(t+1)}\| \leq \frac{\zeta}{\mu^{(t)}} \qquad (\forall i \in [n]).$$

*Proof.* From Line 6 of Algorithm 1, $\mathbf{x}_i^{(t+1)}$ is local solution to the strongly convex subproblem: $\mathbf{x}_i^{(t+1)} = \arg\min_{\mathbf{y}\in\mathcal{X}}(f_i(\mathbf{y}) + \frac{\mu^{(t)}}{2}\|\mathbf{y} - \mathbf{x}^{(t)}\|^2)$. However, this subproblem does not admit a closed-form solution unless $f_i$ is a particularly simple function (e.g., quadratic function). Instead, it is natural to view $x_i^{(t+1)}$ as the limit point of iterative gradient descent applied to the objective above. To formalize this, let us assume a conceptual learning rate $\nu \leq \frac{1}{\mu^{(t)}}$ that is sufficiently small to ensure convergence. Denote by $\{\mathbf{x}_i^{(t,l)}\}_{l=1}^{\infty}$ the gradient descent iterates initialized at $\mathbf{x}_i^{(t,0)} = \mathbf{x}^{(t)}$ with conceptual learning rate $\nu$. Then, $\mathbf{x}_i^{(t+1)}$ coincides with the limit point $\mathbf{x}_i^{(t,\infty)}$. Based on this observation, for any two distinct nodes $i, j$, we have

$$\|\mathbf{x}_i^{(t,l+1)} - \mathbf{x}_j^{(t,l+1)}\|$$

$$\leq \left\|\left(1 - \nu\mu^{(t)}\right)\left(\mathbf{x}_i^{(t,l)} - \mathbf{x}_j^{(t,l)}\right) - \nu\left(\nabla f_i(\mathbf{x}_i^{(t,l)}) - \nabla f_j(\mathbf{x}_j^{(t,l)})\right)\right\|$$

$$\leq \left\|\left(1 - \nu\mu^{(t)}\right)\left(\mathbf{x}_i^{(t,l)} - \mathbf{x}_j^{(t,l)}\right) - \nu\left(\nabla f_i(\mathbf{x}_i^{(t,l)}) - \nabla f_i(\mathbf{x}_j^{(t,l)})\right)\right\| + \nu\left\|\nabla f_i(\mathbf{x}_j^{(t,l)}) - \nabla f_j(\mathbf{x}_j^{(t,l)})\right\|$$

$$\leq \left\|\left(1 - \nu\mu^{(t)} - \nu\nabla^2 f_i(\widetilde{\mathbf{x}}_{i,j}^{(t,l)})\right)\left(\mathbf{x}_i^{(t,l)} - \mathbf{x}_j^{(t,l)}\right)\right\| + \nu\zeta$$

$$\leq \left(1 - \nu\mu^{(t)}\right)\left\|\mathbf{x}_i^{(t,l)} - \mathbf{x}_j^{(t,l)}\right\| + \nu\zeta$$

$$\leq \nu\zeta\sum_{l'=0}^{l}\left(1 - \nu\mu^{(t)}\right)^{l-l'},$$

where $\widetilde{\mathbf{x}}_{i,j}^{(t,l)}$ is a point on the line between $\mathbf{x}_i^{(t,l)}$ and $\mathbf{x}_j^{(t,l)}$ specified by Taylor's theorem. By taking limitation of $l \to \infty$, we obtain

$$\|\mathbf{x}_i^{(t+1)} - \mathbf{x}_j^{(t+1)}\| = \|\mathbf{x}_i^{(t,\infty)} - \mathbf{x}_j^{(t,\infty)}\| \leq \frac{\zeta}{\mu^{(t)}}.$$

From this, we get the statement. $\qquad\square$

**Lemma 11.** *Suppose that Assumptions 1, 2, 4, 5 hold. For the iterations generated by Line 6 of Algorithm 1, we have:*

$$\Delta^{(t+1)} \leq \zeta G.$$

*Proof.*

$$\Delta^{(t+1)} = \mu^{(t)} \cdot \max\left\{f(\mathbf{x}^{(t+1)}) - \frac{1}{n}\sum_{i=1}^{n} f_i(\mathbf{x}_i^{(t+1)}) - \frac{\mu^{(t)}}{2n}\sum_{i=1}^{n}\|\mathbf{x}_i^{(t+1)} - \mathbf{x}^{(t)}\|^2, 0\right\}$$

$$\leq \mu^{(t)} \cdot \max\left\{\frac{1}{n}\sum_{i=1}^{n}\left(f_i(\mathbf{x}^{(t+1)}) - f_i(\mathbf{x}_i^{(t+1)})\right), 0\right\}$$

$$\leq \frac{\mu^{(t)}}{n}\sum_{i=1}^{n}\left|\left\langle\nabla f_i(\mathbf{x}^{(t+1)}), \mathbf{x}^{(t+1)} - \mathbf{x}_i^{(t+1)}\right\rangle\right|$$

$$\leq \frac{\mu^{(t)}}{n}\sum_{i=1}^{n}\left\|\nabla f_i(\mathbf{x}^{(t+1)})\right\|\left\|\mathbf{x}^{(t+1)} - \mathbf{x}_i^{(t+1)}\right\|$$

$$\leq \frac{\mu^{(t)}G}{n}\sum_{i=1}^{n}\left\|\mathbf{x}^{(t+1)} - \mathbf{x}_i^{(t+1)}\right\|,$$

where Assumption 4 is used in the last line. From Lemma 10, we get $\Delta^{(t+1)} \leq \zeta G$. $\qquad\square$

**Lemma 12.** *Suppose that Assumptions 1-5 hold. For the iterations generated by Line 6 of Algorithm 1, we have:*

[**FedProxLoD**]

$$\sum_{k=0}^{T-1} \min\left\{\frac{\mu^{(k)}}{\mu^{(k+1)}}, 1\right\} r^{(k)} \left(f(\mathbf{x}^{(k+1)}) - f(\mathbf{x}^*)\right) \leq 2r^{(T)}\left(\overline{d}^{(T)} + r^{(T)}\right)\sqrt{u^{(0)} + \zeta GT},$$

[**FedProxWLoD**]

$$\sum_{k=0}^{T-1} \min\left\{\frac{\mu^{(k)}}{\mu^{(k+1)}}, 1\right\} (r^{(k)})^2 \left(f(\mathbf{x}^{(k+1)}) - f(\mathbf{x}^*)\right) \leq 2r^{(T)}\left(\overline{d}^{(T)} + r^{(T)}\right)\sqrt{u^{(0)} + (r^{(T)})^2\zeta GT}.$$

*Proof.* Using Lemma 11 and update rule in FedProxLoD: $u^{(t+1)} = u^{(t)} + \Delta^{(t+1)}$,

$$u^{(T)} = u^{(T-1)} + \Delta^{(T)}$$

$$= u^{(0)} + \sum_{k=1}^{T} \Delta^{(k)}$$

$$\leq u^{(0)} + \zeta GT.$$

Meanwhile, for FedProxWLoD using $u^{(t+1)} = u^{(t)} + (r^{(t+1)})^2\Delta^{(t+1)}$, we obtain

$$u^{(T)} = u^{(T-1)} + (r^{(T)})^2\Delta^{(T)}$$

$$= u^{(0)} + \sum_{k=1}^{T} (r^{(k)})^2\Delta^{(k)}$$

$$\leq u^{(0)} + (r^{(T)})^2 \sum_{k=1}^{T} \Delta^{(k)}$$

$$\leq u^{(0)} + (r^{(T)})^2\zeta GT.$$

Integrating these inequalities with Lemma 9 results in the statement. $\qquad\square$

**Lemma 13.** *For the iterations generated by Line 6 of Algorithm 1, we have:*

[**FedProxLoD**]

$$\max_{T'\leq T}\sum_{k=0}^{T'-1} \min\left\{\frac{\mu^{(k)}}{\mu^{(k+1)}}, 1\right\} \frac{r^{(k)}}{r^{(T')}} \geq \max_{T'\leq T}\sum_{k=0}^{T'-1} \frac{r^{(k)}\sqrt{u^{(k)}}}{r^{(T')}\sqrt{u^{(T')}}} \geq \frac{1}{e}\left(\frac{T}{\log_+\left(\frac{r^{(T)}\sqrt{u^{(T)}}}{r^{(0)}\sqrt{u^{(0)}}}\right)} - 1\right),$$

[**FedProxWLoD**]

$$\max_{T'\leq T}\sum_{k=0}^{T'-1} \min\left\{\frac{\mu^{(k)}}{\mu^{(k+1)}}, 1\right\} \left(\frac{r^{(k)}}{r^{(T')}}\right)^2 \geq \max_{T'\leq T}\sum_{k=0}^{T'-1} \left(\frac{r^{(k)}}{r^{(T')}}\right)^2 \frac{\sqrt{u^{(k)}}}{\sqrt{u^{(T')}}} \geq \frac{1}{e}\left(\frac{T}{\log_+\left(\left(\frac{r^{(T)}}{r^{(0)}}\right)^2\frac{\sqrt{u^{(T)}}}{\sqrt{u^{(0)}}}\right)} - 1\right).$$

*Proof.* For FedProxLoD and any $T'$ satisfying $k \leq T' \leq T$, we heve

$$\min\left\{\frac{\mu^{(k)}}{\mu^{(k+1)}}, 1\right\} \frac{r^{(k)}}{r^{(t)}} = \min\left\{\frac{r^{(k+1)}\sqrt{u^{(k)}}}{r^{(k)}\sqrt{u^{(k+1)}}}, 1\right\} \frac{r^{(k)}}{r^{(T')}}$$

$$\geq \frac{r^{(k)}\sqrt{u^{(k)}}}{r^{(T')}\sqrt{u^{(k+1)}}}$$

$$\geq \frac{r^{(k)}\sqrt{u^{(k)}}}{r^{(T')}\sqrt{u^{(T')}}}$$

Meanwhile for FedProxWLoD and any $T'$ satisfying $k \leq T' \leq T$, we heve

$$\min\left\{\frac{\mu^{(k)}}{\mu^{(k+1)}}, 1\right\}\left(\frac{r^{(k)}}{r^{(t)}}\right)^2 = \min\left\{\frac{r^{(k+1)}\sqrt{u^{(k)}}}{r^{(k)}\sqrt{u^{(k+1)}}}, 1\right\}\left(\frac{r^{(k)}}{r^{(T')}}\right)^2$$

$$\geq \left(\frac{r^{(k)}}{r^{(T')}}\right)^2 \frac{\sqrt{u^{(k)}}}{\sqrt{u^{(k+1)}}}$$

$$\geq \left(\frac{r^{(k)}}{r^{(T')}}\right)^2 \frac{\sqrt{u^{(k)}}}{\sqrt{u^{(T')}}}$$

By using Lemma 4, we get the statement. $\qquad\square$

Finally, proof of Theorem 2 is shown.

**Theorem 2** ((Formal) convergence rates of FedProxLoD and FedProxWLoD)**.**

[**FedProxLoD**]

*Suppose that Assumptions 1-5 hold. For the iterations generated by FedProxLoD in Algorithm 1, a certain large $T$ such that satisfies $T \geq 2\log_+\left(\frac{2\sqrt{2}D_0\sqrt{\zeta G T}}{r^{(0)}\sqrt{u^{(0)}}}\right)$, $r^{(0)} \leq 2D_0$, $u^{(0)} \leq \zeta GT$, we have:*

$$\min_{T' \leq T}\left(f(\mathbf{x}_{out}^{(T')}) - f(\mathbf{x}^*)\right) \leq \widetilde{\mathcal{O}}\left(\frac{\sqrt{\zeta G}D_0}{\sqrt{T}}\right),$$

*where $\mathbf{x}_{out}^{(T')} := \frac{1}{\sum_{k=0}^{T'-1}\min\left\{\frac{\mu^{(k)}}{\mu^{(k+1)}}, 1\right\}r^{(k)}}\sum_{k=0}^{T'-1}\min\left\{\frac{\mu^{(k)}}{\mu^{(k+1)}}, 1\right\}r^{(k)}\mathbf{x}^{(k+1)}$. When introducing $\mathbf{x}_{best}^{(T)} := \arg\min_{\mathbf{x} \in \{\mathbf{x}^{(T')}\}_{T' \in [T]}} f(\mathbf{x})$, we have*

$$f(\mathbf{x}_{best}^{(T)}) - f(\mathbf{x}^*) \leq \widetilde{\mathcal{O}}\left(\frac{\sqrt{\zeta G}D_0}{\sqrt{T}}\right).$$

[**FedProxWLoD**]

*Suppose that Assumptions 1-5 hold. For the iterations generated by FedProxWLoD in Algorithm 1, a certain large $T$ such that satisfies $T \geq 2\log_+\left(\frac{2\sqrt{3}(D_0)^3\sqrt{\zeta G T}}{(r^{(0)})^2\sqrt{u^{(0)}}}\right)$, $r^{(0)} \leq 2D_0$, $u^{(0)} \leq 2(r^{(0)})^2\zeta GT$, we have:*

$$\min_{T' \leq T}\left(f(\mathbf{x}_{out}^{(T')}) - f(\mathbf{x}^*)\right) \leq \widetilde{\mathcal{O}}\left(\frac{\sqrt{\zeta G}D_0}{\sqrt{T}}\right),$$

*where $\mathbf{x}_{out}^{(T')} := \frac{1}{\sum_{k=0}^{T'-1}\min\left\{\frac{\mu^{(k)}}{\mu^{(k+1)}}, 1\right\}(r^{(k)})^2}\sum_{k=0}^{T'-1}\min\left\{\frac{\mu^{(k)}}{\mu^{(k+1)}}, 1\right\}(r^{(k)})^2\mathbf{x}^{(k+1)}$. When introducing $\mathbf{x}_{best}^{(T)} := \arg\min_{\mathbf{x} \in \{\mathbf{x}^{(T')}\}_{T' \in [T]}} f(\mathbf{x})$, we have*

$$f(\mathbf{x}_{best}^{(T)}) - f(\mathbf{x}^*) \leq \widetilde{\mathcal{O}}\left(\frac{\sqrt{\zeta G}D_0}{\sqrt{T}}\right).$$

*Proof.* First, the convergence rate of FedProxLoD is derived. From Lemma 12, we obtain

$$\frac{1}{\sum_{k=0}^{T'-1}\min\left\{\frac{\mu^{(k)}}{\mu^{(k+1)}}, 1\right\}\frac{r^{(k)}}{r^{(T')}}}\sum_{k=0}^{T'-1}\min\left\{\frac{\mu^{(k)}}{\mu^{(k+1)}}, 1\right\}\frac{r^{(k)}}{r^{(T')}}\left(f(\mathbf{x}^{(k+1)}) - f(\mathbf{x}^*)\right)$$

$$\leq \frac{2(\overline{d}^{(T')} + r^{(T')})\sqrt{u^{(0)} + \zeta GT'}}{\sum_{k=0}^{T'-1}\min\left\{\frac{\mu^{(k)}}{\mu^{(k+1)}}, 1\right\}\frac{r^{(k)}}{r^{(T')}}},$$

for any $T' \leq T$. Using Lemma 13, we get

$$\min_{T' \leq T} \frac{1}{\sum_{k=0}^{T'-1} \min\left\{\frac{\mu^{(k)}}{\mu^{(k+1)}}, 1\right\} \frac{r^{(k)}}{r^{(T')}}} \sum_{k=0}^{T'-1} \min\left\{\frac{\mu^{(k)}}{\mu^{(k+1)}}, 1\right\} \frac{r^{(k)}}{r^{(T')}} \left(f(\mathbf{x}^{(k+1)}) - f(\mathbf{x}^*)\right)$$

$$\leq \frac{2\left(\overline{d}^{(T)} + r^{(T)}\right)\sqrt{u^{(0)} + \zeta GT}}{\max_{T' \leq T} \sum_{k=0}^{T'-1} \min\left\{\frac{\mu^{(k)}}{\mu^{(k+1)}}, 1\right\} \frac{r^{(k)}}{r^{(T')}}}$$

$$\leq \frac{2\left(\overline{d}^{(T)} + r^{(T)}\right)\sqrt{u^{(0)} + \zeta GT}}{\frac{1}{e}\left(\frac{T}{\log_+\left(\frac{r^{(T)}\sqrt{u^{(T)}}}{r^{(0)}\sqrt{u^{(0)}}}\right)} - 1\right)}$$

$$\leq \frac{2\left(\overline{d}^{(T)} + r^{(T)}\right)\sqrt{u^{(0)} + \zeta GT}}{\frac{1}{e}\left(\frac{T}{\log_+\left(\frac{r^{(T)}\sqrt{u^{(0)} + \zeta GT}}{r^{(0)}\sqrt{u^{(0)}}}\right)} - 1\right)}.$$

For a certain large $T'$ such that satisfies $T' \geq 2\log_+\left(\frac{r^{(T)}\sqrt{u^{(0)} + \zeta GT}}{r^{(0)}\sqrt{u^{(0)}}}\right)$, i.e., $\frac{T'}{\log_+\left(\frac{r^{(T)}\sqrt{u^{(0)} + \zeta GT}}{r^{(0)}\sqrt{u^{(0)}}}\right)} - 1 \geq \frac{1}{2} \cdot \frac{T'}{\log_+\left(\frac{r^{(T)}\sqrt{u^{(0)} + \zeta GT}}{r^{(0)}\sqrt{u^{(0)}}}\right)}$, we get

$$\min_{T' \leq T} \frac{1}{\sum_{k=0}^{T'-1} \min\left\{\frac{\mu^{(k)}}{\mu^{(k+1)}}, 1\right\} \frac{r^{(k)}}{r^{(T')}}} \sum_{k=0}^{T'-1} \min\left\{\frac{\mu^{(k)}}{\mu^{(k+1)}}, 1\right\} \frac{r^{(k)}}{r^{(T')}} \left(f(\mathbf{x}^{(k+1)}) - f(\mathbf{x}^*)\right)$$

$$\leq \frac{4e\left(\overline{d}^{(T)} + r^{(T)}\right)\sqrt{u^{(0)} + \zeta GT}}{T} \log_+\left(\frac{r^{(T)}\sqrt{u^{(0)} + \zeta GT}}{r^{(0)}\sqrt{u^{(0)}}}\right)$$

$$\leq \frac{12\sqrt{2}eD_0\sqrt{\zeta G}}{\sqrt{T}} \log_+\left(\frac{2\sqrt{2}D_0\sqrt{\zeta GT}}{r^{(0)}\sqrt{u^{(0)}}}\right)$$

$$= \widetilde{\mathcal{O}}\left(\frac{D_0\sqrt{\zeta G}}{\sqrt{T}}\right),$$

where we used $\overline{d}^{(t)} = \max_{k \leq t} d^{(k)} \leq D_0$ since $d^{(t+1)} \leq d^{(t)}$, $r^{(T)} \leq 2D_0$, and $u^{(0)} \leq \zeta GT$ in the second inequality. In the last line, the logarithmic term is omitted under big-O notation. Since the initial values $r^{(0)}, u^{(0)}$ appear inside the logarithmic term, this omission implicitly reflects the sensitivity to these initial values.

In addition, the LHS term can be evaluated as follows:

$$\min_{T' \leq T} \frac{1}{\sum_{k=0}^{T'-1} \min\left\{\frac{\mu^{(k)}}{\mu^{(k+1)}}, 1\right\} \frac{r^{(k)}}{r^{(T')}}} \sum_{k=0}^{T'-1} \min\left\{\frac{\mu^{(k)}}{\mu^{(k+1)}}, 1\right\} \frac{r^{(k)}}{r^{(T')}} \left(f(\mathbf{x}^{(k+1)}) - f(\mathbf{x}^*)\right)$$

$$\geq \min_{T' \leq T} \left(f(\mathbf{x}_{\text{out}}^{(T')}) - f(\mathbf{x}^*)\right)$$

where $\mathbf{x}_{\text{out}}^{(T')} := \frac{1}{\sum_{k=0}^{T'-1} \min\left\{\frac{\mu^{(k)}}{\mu^{(k+1)}}, 1\right\} r^{(k)}} \sum_{k=0}^{T'-1} \min\left\{\frac{\mu^{(k)}}{\mu^{(k+1)}}, 1\right\} r^{(k)}\mathbf{x}^{(k+1)}$. Particularly, when we denote $\mathbf{x}_{\text{best}}^{(T)} := \arg\min_{\mathbf{x} \in \{\mathbf{x}^{(T')}\}_{T' \in [T]}} f(\mathbf{x})$, we get

$$f(\mathbf{x}_{\text{best}}^{(T)}) - f(\mathbf{x}^*) \leq \widetilde{\mathcal{O}}\left(\frac{D_0\sqrt{\zeta G}}{\sqrt{T}}\right).$$

Next, the convergence rate for FedProxWLoD is given. From Lemma 12 and using $r^{(T)} \leq 2D_0$ and $u^{(0)} \leq 2(r^{(0)})^2 \zeta GT$, we obtain

$$
\frac{1}{\sum_{k=0}^{T'-1} \min\left\{\frac{\mu^{(k)}}{\mu^{(k+1)}}, 1\right\} \left(\frac{r^{(k)}}{r^{(T')}}\right)^2} \sum_{k=0}^{T'-1} \min\left\{\frac{\mu^{(k)}}{\mu^{(k+1)}}, 1\right\} \left(\frac{r^{(k)}}{r^{(T')}}\right)^2 \left(f(\mathbf{x}^{(k+1)}) - f(\mathbf{x}^*)\right)
$$

$$
\leq \frac{2\sqrt{3}\left(\overline{d}^{(T')} + r^{(T')}\right)\sqrt{\zeta GT'}}{\sum_{k=0}^{T'-1} \min\left\{\frac{\mu^{(k)}}{\mu^{(k+1)}}, 1\right\} \left(\frac{r^{(k)}}{r^{(T')}}\right)^2}.
$$

for any $T' \leq T$. Using Lemma 13, we obtain

$$
\min_{T' \leq T} \frac{1}{\sum_{k=0}^{T'-1} \min\left\{\frac{\mu^{(k)}}{\mu^{(k+1)}}, 1\right\} \left(\frac{r^{(k)}}{r^{(T')}}\right)^2} \sum_{k=0}^{T'-1} \min\left\{\frac{\mu^{(k)}}{\mu^{(k+1)}}, 1\right\} \left(\frac{r^{(k)}}{r^{(T')}}\right)^2 \left(f(\mathbf{x}^{(k+1)}) - f(\mathbf{x}^*)\right)
$$

$$
\leq \frac{2\sqrt{3}\left(\overline{d}^{(T)} + r^{(T)}\right)\sqrt{\zeta GT}}{\max_{T' \leq T} \sum_{k=0}^{T'-1} \min\left\{\frac{\mu^{(k)}}{\mu^{(k+1)}}, 1\right\} \left(\frac{r^{(k)}}{r^{(T')}}\right)^2}
$$

$$
\leq \frac{2\sqrt{3}\left(\overline{d}^{(T)} + r^{(T)}\right)\sqrt{\zeta GT}}{\frac{1}{e}\left(\frac{T}{\log_+\left(\left(\frac{r^{(T)}}{r^{(0)}}\right)^2 \frac{\sqrt{u^{(T)}}}{\sqrt{u^{(0)}}}\right)} - 1\right)}
$$

$$
\leq \frac{2\sqrt{3}\left(\overline{d}^{(T)} + r^{(T)}\right)\sqrt{\zeta GT}}{\frac{1}{e}\left(\frac{T}{\log_+\left(\left(\frac{r^{(T)}}{r^{(0)}}\right)^2 \frac{\sqrt{u^{(0)} + (r^{(T)})^2 \zeta GT}}{\sqrt{u^{(0)}}}\right)} - 1\right)}.
$$

For a certain large $T$ such that satisfies $T \geq 2\log_+\left(\left(\frac{r^{(T)}}{r^{(0)}}\right)^2 \frac{\sqrt{u^{(0)} + (r^{(T)})^2 \zeta GT}}{\sqrt{u^{(0)}}}\right)$, i.e.,

$$
\frac{T}{\log_+\left(\left(\frac{r^{(T)}}{r^{(0)}}\right)^2 \frac{\sqrt{u^{(0)} + (r^{(T)})^2 \zeta GT}}{\sqrt{u^{(0)}}}\right)} - 1 \geq \frac{1}{2} \cdot \frac{T}{\log_+\left(\left(\frac{r^{(T)}}{r^{(0)}}\right)^2 \frac{\sqrt{u^{(0)} + (r^{(T)})^2 \zeta GT}}{\sqrt{u^{(0)}}}\right)}.
$$

$$
\min_{T' \leq T} \frac{1}{\sum_{k=0}^{T'-1} \min\left\{\frac{\mu^{(k)}}{\mu^{(k+1)}}, 1\right\} \left(\frac{r^{(k)}}{r^{(T')}}\right)^2} \sum_{k=0}^{T'-1} \min\left\{\frac{\mu^{(k)}}{\mu^{(k+1)}}, 1\right\} \left(\frac{r^{(k)}}{r^{(T')}}\right)^2 \left(f(\mathbf{x}^{(k+1)}) - f(\mathbf{x}^*)\right)
$$

$$
\leq \frac{4\sqrt{3}e\left(\overline{d}^{(T)} + r^{(T)}\right)\sqrt{\zeta GT}}{T} \log_+\left(\left(\frac{r^{(T)}}{r^{(0)}}\right)^2 \frac{\sqrt{u^{(0)} + (r^{(T)})^2 \zeta GT}}{\sqrt{u^{(0)}}}\right)
$$

$$
\leq \frac{12\sqrt{3}e D_0 \sqrt{\zeta G}}{\sqrt{T}} \log_+\left(\frac{\sqrt{3}(r^{(T)})^3 \sqrt{\zeta GT}}{(r^{(0)})^2 \sqrt{u^{(0)}}}\right)
$$

$$
= \widetilde{\mathcal{O}}\left(\frac{D_0 \sqrt{\zeta G}}{\sqrt{T}}\right),
$$

where used $\overline{d}^{(t)} = \max_{k \leq t} d^{(k)} \leq D_0$ since $d^{(t+1)} \leq d^{(t)}$, $r^{(T)} \leq 2D_0$, and $u^{(0)} \leq 2(r^{(0)})^2 \zeta GT$ in the second inequality. In addition, the LHS term can be evaluated as follows:

$$
\min_{T' \leq T} \frac{1}{\sum_{k=0}^{T'-1} \min\left\{\frac{\mu^{(k)}}{\mu^{(k+1)}}, 1\right\} \left(\frac{r^{(k)}}{r^{(T')}}\right)^2} \sum_{k=0}^{T'-1} \min\left\{\frac{\mu^{(k)}}{\mu^{(k+1)}}, 1\right\} \left(\frac{r^{(k)}}{r^{(T')}}\right)^2 \left(f(\mathbf{x}^{(k+1)}) - f(\mathbf{x}^*)\right)
$$

$$
\geq \min_{T' \leq T} \left(f(\mathbf{x}_{\text{out}}^{(T')}) - f(\mathbf{x}^*)\right)
$$

where $\mathbf{x}_{\text{out}}^{(T')} := \frac{1}{\sum_{k=0}^{T'-1} \min\left\{\frac{\mu^{(k)}}{\mu^{(k+1)}},1\right\}(r^{(k)})^2} \sum_{k=0}^{T'-1} \min\left\{\frac{\mu^{(k)}}{\mu^{(k+1)}},1\right\}(r^{(k)})^2 \mathbf{x}^{(k+1)}$. When we de-

note $\mathbf{x}_{\text{best}}^{(T)} := \arg\min_{\mathbf{x} \in \{\mathbf{x}^{(T')}\}_{T' \in [T]}} f(\mathbf{x})$, we get

$$f(\mathbf{x}_{\text{best}}^{(T)}) - f(\mathbf{x}^*) \leq \widetilde{\mathcal{O}}\Big(\frac{D_0\sqrt{\zeta G}}{\sqrt{T}}\Big).$$

$\square$

## C.2 PROOFS FOR REMARK 2

**Lemma 14.** *Suppose that Assumption 5 holds. Then, $(\sqrt{2}\zeta, \sqrt{2})$-BGD holds.*

*Proof.* $\|\nabla f_i(x)\| \leq \|\nabla f_i(x) - \nabla f(x)\| + \|\nabla f(x)\|$. From this, we get $\|\nabla f_i(x)\|^2 \leq 2\|\nabla f_i(x) - \nabla f(x)\|^2 + 2\|\nabla f(x)\|^2 \leq 2\zeta^2 + 2\|\nabla f(x)\|^2$, which is $(\sqrt{2}\zeta, \sqrt{2})$-BGD. $\square$

# D IMPLEMENTATION OF PROPOSED ALGORITHMS

Our FedProxLoD and FedProxWLoD are formalized in Algorithm 1. However, this algorithm is not directly applicable to complex models, such as deep neural networks, due to the computational difficulty of exactly solving the local subproblems. To address this, we employ the approximated updated rules, presented in Algorithm 3. The additional operations introduced for this approximation are highlighted in blue. To adaptively determine learning rate $\eta^{(t)}$, the learning rate formulation of DoG and DoWG in (2) is simply extended to use averaged local gradients over $n$ clients. In our experimental results in Section 5, we used $K = 100$. Additionally, in Line 24 of Algorithm 3, the central server requires access to a dataset to compute the global loss using the global model $f(\mathbf{x}^{(t+1)})$. To enable this, $1,000$ data samples are homogeneously picked from the dataset, and all $1,000$ samples are used to compute $f(\mathbf{x}^{(t+1)})$.

---

**Algorithm 3** FedProxLoD and FedProxWLoD implementation used in our experiments in Section 5

---

1: Initialization $\mathbf{x}^{(0)} = \mathbf{x}_{\text{out}}^{(0)} = \mathbf{x}_{\text{best}}^{(0)}, r^{(0)}(> 0), u^{(0)}(> 0), v^{(0)}(> 0), w_2^{(0)} = 0$
2: **if** (FedProxLoD) $\mu^{(0)} = \sqrt{u^{(0)}}/r^{(0)}, \eta^{(0)} = r^{(0)}/\sqrt{v^{(0)}},$
3: **else if** (FedProxWLoD) $\mu^{(0)} = \sqrt{u^{(0)}}/(r^{(0)})^2, \eta^{(0)} = (r^{(0)})^2/\sqrt{v^{(0)}}$ **end**
4: **for** $t = 0, 1, \ldots, T - 1$ **do**
5:   ▷ Client procedure
6:   **for** $i = 1, \ldots, n$ **do**
7:     $\boldsymbol{x}_i^{(t,0)} = \mathbf{x}_{\text{best}}^{(t)}$
8:     **for** $k = 0, \ldots, K - 1$ **do**
9:       $\xi_i^{(t,k)} \sim \mathcal{D}_i$
10:       $\boldsymbol{x}_i^{(t,k+1)} = \boldsymbol{x}_i^{(t,k)} - \eta^{(t)} \big( \nabla f_i(\boldsymbol{x}_i^{(t,k)}; \xi_i^{(t,k)}) + \mu^{(t)}(\mathbf{x}_{\text{best}}^{(t)} - \boldsymbol{x}_i^{(t,k)}) \big)$
11:     **end for**
12:     $\mathbf{x}_i^{(t+1)} = \boldsymbol{x}_i^{(t,K)}$
13:   **end for**
14:   $\mathbf{Transmit}_{\text{Client } i \to \text{Server}}(\mathbf{x}_i^{(t+1)}, f_i(\mathbf{x}_i^{(t+1)}), \|\nabla f_i(\mathbf{x}_i^{(t+1)})\|^2)$
15:   ▷ Server procedure
16:   $\mathbf{x}^{(t+1)} = \frac{1}{n} \sum_{i=1}^n \mathbf{x}_i^{(t+1)}$
17:   $r^{(t+1)} = \max\{\|\mathbf{x}^{(t+1)} - \mathbf{x}^{(0)}\|, r^{(t)}\}$
18:   $\Delta^{(t+1)} = \mu^{(t)} \cdot \max\{f(\mathbf{x}^{(t+1)}) - \frac{1}{n} \sum_{i=1}^n f_i(\mathbf{x}_i^{(t+1)}) - \frac{\mu^{(t)}}{2n} \sum_{i=1}^n \|\mathbf{x}_i^{(t+1)} - \mathbf{x}^{(t)}\|^2, 0\}$
19:   **if** (FedProxLoD) **then**
20:     $u^{(t+1)} = u^{(t)} + \Delta^{(r+1)}$
21:     $\mu^{(t+1)} = \sqrt{u^{(t+1)}}/r^{(t+1)}$
22:     $v^{(t+1)} = v^{(t)} + \frac{1}{n} \sum_{i=1}^n \|\nabla f_i(\mathbf{x}_i^{(t+1)})\|^2$
23:     $\eta^{(t+1)} = r^{(t+1)}/\sqrt{v^{(t+1)}}$
24:     $w_1^{(t+1)} = \min\{\frac{\mu^{(t+1)}}{\mu^{(t)}}, 1\} r^{(t+1)}$
25:   **else if** (FedProxWLoD) **then**
26:     $u^{(t+1)} = u^{(t)} + (r^{(t+1)})^2 \Delta^{(r+1)}$
27:     $\mu^{(t+1)} = \sqrt{u^{(t+1)}}/(r^{(t+1)})^2$
28:     $v^{(t+1)} = v^{(t)} + \frac{1}{n} \sum_{i=1}^n (r^{(t+1)})^2 \|\nabla f_i(\mathbf{x}_i^{(t+1)})\|^2$
29:     $\eta^{(t+1)} = (r^{(t+1)})^2/\sqrt{v^{(t+1)}}$
30:     $w_1^{(t+1)} = \min\{\frac{\mu^{(t+1)}}{\mu^{(t)}}, 1\} (r^{(t+1)})^2$
31:   **end if**
32:   $w_2^{(t+1)} = w_2^{(t)} + w_1^{(t+1)}$
33:   $\mathbf{x}_{\text{out}}^{(t+1)} = \frac{1}{w_2^{(t+1)}} \big( w_2^{(t)} \mathbf{x}_{\text{out}}^{(t)} + w_1^{(t+1)} \mathbf{x}^{(t+1)} \big)$
34:   $\mathbf{x}_{\text{best}}^{(t+1)} = \arg\min_{\mathbf{x} \in \{\mathbf{x}_{\text{out}}^{(t+1)}, \mathbf{x}_{\text{best}}^{(t)}\}} f(\mathbf{x})$
35:   $\mathbf{Transmit}_{\text{Server} \to n \text{ clients}}(\mathbf{x}_{\text{best}}^{(t+1)}, \mu^{(t+1)}, \eta^{(t+1)})$
36: **end for**

---

Although Algorithm 3 was employed in Section 5, there are scenarios in which holding datasets on a central server is not feasible. In such cases, Algorithm 4, which requires additional communication as highlighted in red, can be used instead.

---

**Algorithm 4** Another implementation of FedProxLoD and FedProxWLoD

---

1: Initialization $\mathbf{x}^{(0)} = \mathbf{x}_{\text{out}}^{(0)} = \mathbf{x}_{\text{best}}^{(0)}, r^{(0)}(> 0), u^{(0)}(> 0), v^{(0)}(> 0), w_2^{(0)} = 0$

2: **if** (FedProxLoD) $\mu^{(0)} = \sqrt{u^{(0)}}/r^{(0)}, \eta^{(0)} = r^{(0)}/\sqrt{v^{(0)}},$

3: **else if** (FedProxWLoD) $\mu^{(0)} = \sqrt{u^{(0)}}/(r^{(0)})^2, \eta^{(0)} = (r^{(0)})^2/\sqrt{v^{(0)}}$ **end**

4: **for** $t = 0, 1, \ldots, T-1$ **do**

5:    ▷ Client procedure

6:    **for** $i = 1, \ldots, n$ **do**

7:      $\boldsymbol{x}_i^{(t,0)} = \mathbf{x}_{\text{best}}^{(t)}$

8:      **for** $k = 0, \ldots, K-1$ **do**

9:        $\xi_i^{(t,k)} \sim \mathcal{D}_i$

10:        $\boldsymbol{x}_i^{(t,k+1)} = \boldsymbol{x}_i^{(t,k)} - \eta^{(t)}\big(\nabla f_i(\boldsymbol{x}_i^{(t,k)}; \xi_i^{(t,k)}) + \mu^{(t)}(\mathbf{x}_{\text{best}}^{(t)} - \boldsymbol{x}_i^{(t,k)})\big)$

11:      **end for**

12:      $\mathbf{x}_i^{(t+1)} = \boldsymbol{x}_i^{(t,K)}$

13:    **end for**

14:    $\mathbf{Transmit}_{\text{Client}\,i \to \text{Server}}(\mathbf{x}_i^{(t+1)}, f_i(\mathbf{x}_i^{(t+1)}), \|\nabla f_i(\mathbf{x}_i^{(t+1)})\|^2)$

15:    ▷ Server procedure

16:    $\mathbf{x}^{(t+1)} = \frac{1}{n}\sum_{i=1}^n \mathbf{x}_i^{(t+1)}$

17:    ▷ Client procedure

18:    $\mathbf{Transmit}_{\text{Server} \to \text{Client}\,i}(\mathbf{x}^{(t+1)})$

19:    Compute local loss using averaged model $f_i(\mathbf{x}^{(t+1)})$ for each client $i$

20:    $\mathbf{Transmit}_{\text{Client}\,i \to \text{Server}}(f_i(\mathbf{x}^{(t+1)}))$

21:    ▷ Server procedure

22:    Compute global loss $f(\mathbf{x}^{(t+1)}) = \frac{1}{n}\sum_{i=1}^n f_i(\mathbf{x}^{(t+1)})$

23:    $r^{(t+1)} = \max\{\|\mathbf{x}^{(t+1)} - \mathbf{x}^{(0)}\|, r^{(t)}\}$

24:    $\Delta^{(t+1)} = \mu^{(t)} \cdot \max\{f(\mathbf{x}^{(t+1)}) - \frac{1}{n}\sum_{i=1}^n f_i(\mathbf{x}_i^{(t+1)}) - \frac{\mu^{(t)}}{2n}\sum_{i=1}^n \|\mathbf{x}_i^{(t+1)} - \mathbf{x}^{(t)}\|^2, 0\}$

25:    **if** (FedProxLoD) **then**

26:      $u^{(t+1)} = u^{(t)} + \Delta^{(r+1)}$

27:      $\mu^{(t+1)} = \sqrt{u^{(t+1)}}/r^{(t+1)}$

28:      $v^{(t+1)} = v^{(t)} + \frac{1}{n}\sum_{i=1}^n \|\nabla f_i(\mathbf{x}_i^{(t+1)})\|^2$

29:      $\eta^{(t+1)} = r^{(t+1)}/\sqrt{v^{(t+1)}}$

30:      $w_1^{(t+1)} = \min\{\frac{\mu^{(t+1)}}{\mu^{(t)}}, 1\}r^{(t+1)}$

31:    **else if** (FedProxWLoD) **then**

32:      $u^{(t+1)} = u^{(t)} + (r^{(t+1)})^2 \Delta^{(r+1)}$

33:      $\mu^{(t+1)} = \sqrt{u^{(t+1)}}/(r^{(t+1)})^2$

34:      $v^{(t+1)} = v^{(t)} + \frac{1}{n}\sum_{i=1}^n (r^{(t+1)})^2\|\nabla f_i(\mathbf{x}_i^{(t+1)})\|^2$

35:      $\eta^{(t+1)} = (r^{(t+1)})^2/\sqrt{v^{(t+1)}}$

36:      $w_1^{(t+1)} = \min\{\frac{\mu^{(t+1)}}{\mu^{(t)}}, 1\}(r^{(t+1)})^2$

37:    **end if**

38:    $w_2^{(t+1)} = w_2^{(t)} + w_1^{(t+1)}$

39:    $\mathbf{x}_{\text{out}}^{(t+1)} = \frac{1}{w_2^{(t+1)}}\big(w_2^{(t)}\mathbf{x}_{\text{out}}^{(t)} + w_1^{(t+1)}\mathbf{x}^{(t+1)}\big)$

40:    $\mathbf{x}_{\text{best}}^{(t+1)} = \arg\min_{\mathbf{x} \in \{\mathbf{x}_{\text{out}}^{(t+1)}, \mathbf{x}_{\text{best}}^{(t)}\}} f(\mathbf{x})$

41:    $\mathbf{Transmit}_{\text{Server} \to n\text{ clients}}(\mathbf{x}_{\text{best}}^{(t+1)}, \mu^{(t+1)}, \eta^{(t+1)})$

42: **end for**

---

# E    ADDITIONAL EXPERIMENTS

**Computing resource**    We used computing servers employing 8 GPUs (NVIDIA RTX 6000 Ada (48 GB)) and 2 CPUs (AMD EPYC 9354, 3.25 GHz, 32-Core Processor).

**Additional experimental results**    Additional experiments complementing Section 5 are summarized. We evaluate three additional $(n, \alpha)$ settings: $(15, 0.1)$, $(7, 1)$, and $(7, 0.1)$. In line with the main paper, we present data distributions, convergence curves, parameter trajectories, as well as the best and last test accuracies for each scenario.

As noted in Section 5, our proposed FedProxWLod showed competitive performance compared to pre-tuned FL algorithms (e.g., SCAFFOLD). Since FL requires substantial computational resources, achieving strong performance without the need for parameter pre-tuning is a significant advantage.

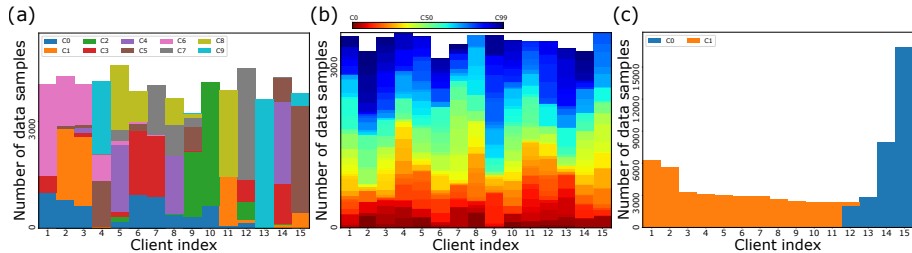

Figure 4: Data distributions using $n = 15$ and $\alpha = 0.1$: (a) fMNIST classification in (T1), (b) CIFAR-100 classification in (T2), and (c) SST-2 classification in (T3).

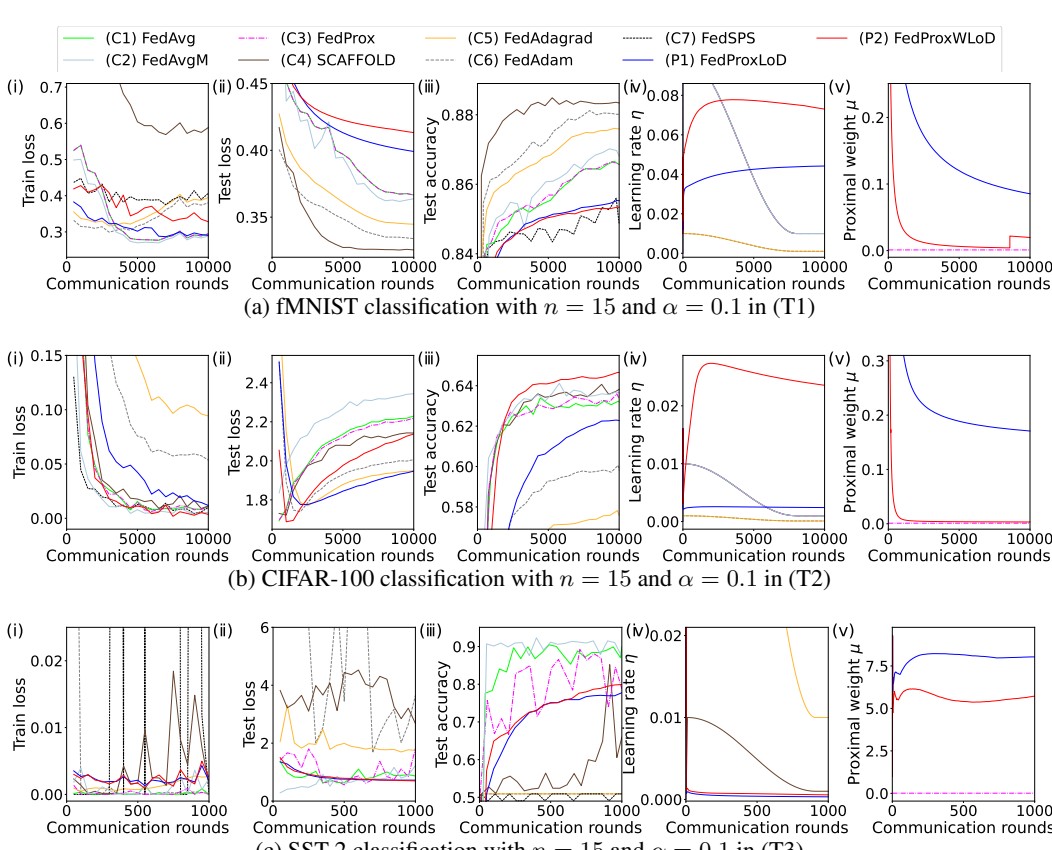

(a) fMNIST classification with $n = 15$ and $\alpha = 0.1$ in (T1)

(b) CIFAR-100 classification with $n = 15$ and $\alpha = 0.1$ in (T2)

(c) SST-2 classification with $n = 15$ and $\alpha = 0.1$ in (T3)

Figure 5: Convergence curves illustrating (i) train loss, (ii) test loss, and (iii) test accuracy, and the evolution of (iv) learning rate and (v) proximal weight when $n = 15$ and $\alpha = 0.1$.

Table 3: Best test accuracy under $n = 15$ and $\alpha = 0.1$. In the comparing algorithms (C1)-(C7), pre-tuning of parameters was conducted. Despite not requiring parameter pre-tuning, our parameter-free algorithms (P1), (P2), (P1'), (P2') achieved competing performance relative to the best performance of pre-tuned baseline algorithms (C1)-(C7).

| Algorithms | Parameters | (T1) Convex-fMNIST | | (T2) ResNet-18-CIFAR-100 | | (T3) BERT-SST-2 | |
|---|---|---|---|---|---|---|---|
| | | Best test acc. | Last test acc. | Best test acc. | Last test acc. | Best test acc. | Last test acc. |
| **(C1) FedAvg** McMahan et al. (2017) | $\{1, 0.1, 0.01, 0.001\} \in \eta$ | 0.8666 | 0.8657 | 0.6339 | 0.6320 | 0.9048 | **0.8990** |
| **(C2) FedAvgM** Hsu et al. (2019) | $\{1, 0.1, 0.01, 0.001\} \in \eta$ | 0.8706 | 0.8704 | 0.6412 | 0.6365 | **0.9232** | 0.8739 |
| **(C3) FedProx** Li et al. (2020) | $\{1, 0.1, 0.01, 0.001\} \in \eta$ $\{0.1, 0.01, 0.001\} \in \mu$ | 0.8666 | 0.8657 | 0.6360 | 0.6336 | 0.9128 | 0.8635 |
| **(C4) SCAFFOLD** Karimireddy et al. (2020) | $\{1, 0.1, 0.01, 0.001\} \in \eta$ | **0.8877** | **0.8829** | 0.6627 | 0.6583 | 0.8521 | 0.6594 |
| **(C5) FedAdaGrad** Reddi et al. (2020) | $\{1, 0.1, 0.01, 0.001\} \in \eta$ | 0.8761 | 0.8759 | 0.6405 | 0.6347 | 0.7615 | 0.7041 |
| **(C6) FedAdam** Reddi et al. (2020) | $\{1, 0.1, 0.01, 0.001\} \in \eta$ | 0.8812 | 0.8803 | **0.6645** | **0.6595** | 0.7901 | 0.7041 |
| **(C7) FedSPS** Mukherjee et al. (2023) | $\{1, 0.1\} \in c$ | 0.8642 | 0.8598 | 0.5550 | 0.5523 | 0.7695 | 0.6594 |
| **(P1) FedProxLoD** | Parameter-free | 0.8550 | 0.8550 | 0.6230 | 0.6228 | 0.7775 | 0.7775 |
| **(P2) FedProxWLoD** | Parameter-free | 0.8534 | 0.8534 | 0.6466 | 0.6466 | **0.7993** | **0.7993** |
| **(P1') (P1) w/o model merge** | Parameter-free | **0.8553** | **0.8553** | 0.6221 | 0.6217 | 0.7672 | 0.7661 |
| **(P2') (P2) w/o model merge** | Parameter-free | 0.8534 | 0.8534 | **0.6475** | **0.6473** | 0.7672 | 0.7672 |

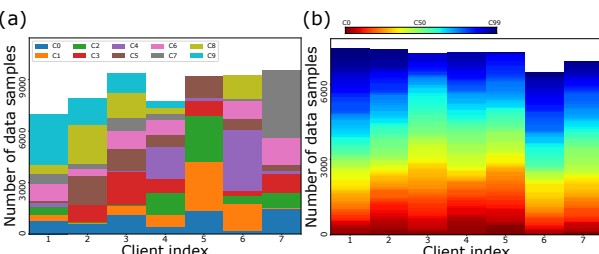

Figure 6: Data distributions using $n = 7$ and $\alpha = 1$: (a) fMNIST classification in (T1), (b) CIFAR-100 classification in (T2), and (c) SST-2 classification in (T3).

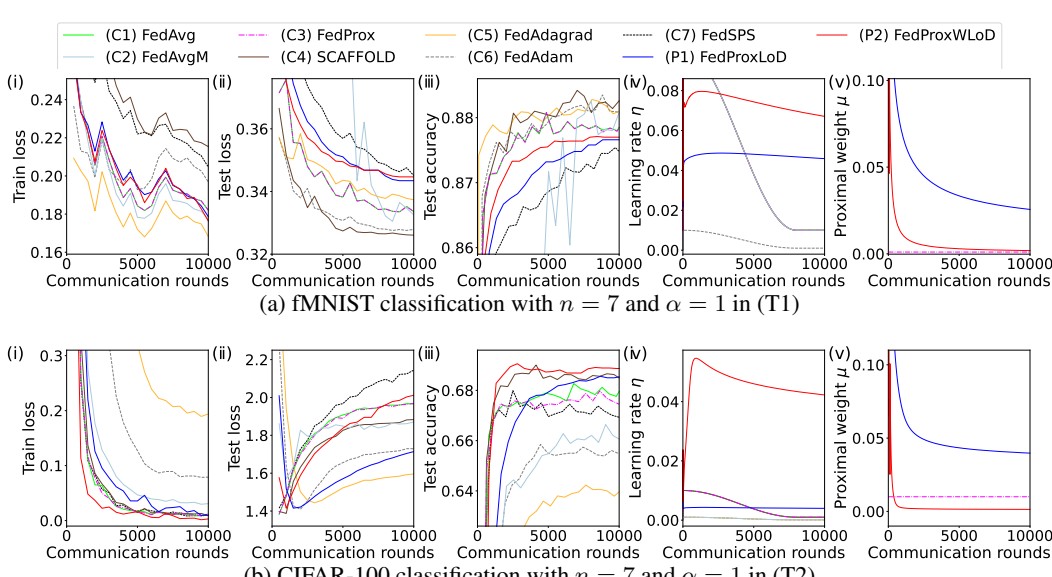

Figure 7: Convergence curves illustrating (i) train loss, (ii) test loss, and (iii) test accuracy, and the evolution of (iv) learning rate and (v) proximal weight when $n = 7$ and $\alpha = 1$.

Table 4: Best test accuracy under $n = 7$ and $\alpha = 1$. In the comparing algorithms (C1)-(C7), pre-tuning of parameters was conducted. Despite not requiring parameter pre-tuning, our parameter-free algorithms (P1), (P2), (P1'), (P2') achieved competing performance relative to the best performance of pre-tuned baseline algorithms (C1)-(C7).

| Algorithms | Parameters | (T1) Convex-fMNIST | | (T2) ResNet-18-CIFAR-100 | |
|---|---|---|---|---|---|
| | | Best test acc. | Last test acc. | Best test acc. | Last test acc. |
| **(C1) FedAvg** McMahan et al. (2017) | $\{1, 0.1, 0.01, 0.001\} \in \eta$ | 0.8807 | 0.8785 | 0.6827 | 0.6796 |
| **(C2) FedAvgM** Hsu et al. (2019) | $\{1, 0.1, 0.01, 0.001\} \in \eta$ | 0.8829 | 0.8808 | 0.6823 | 0.6763 |
| **(C3) FedProx** Li et al. (2020) | $\{1, 0.1, 0.01, 0.001\} \in \eta$ $\{0.1, 0.01, 0.001\} \in \mu$ | 0.8808 | 0.8785 | 0.6800 | 0.6762 |
| **(C4) SCAFFOLD** Karimireddy et al. (2020) | $\{1, 0.1, 0.01, 0.001\} \in \eta$ | **0.8842** | **0.8826** | **0.6901** | **0.6849** |
| **(C5) FedAdaGrad** Reddi et al. (2020) | $\{1, 0.1, 0.01, 0.001\} \in \eta$ | 0.8828 | 0.8810 | 0.6506 | 0.6490 |
| **(C6) FedAdam** Reddi et al. (2020) | $\{1, 0.1, 0.01, 0.001\} \in \eta$ | 0.8835 | 0.8808 | 0.6585 | 0.6551 |
| **(C7) FedSPS** Mukherjee et al. (2023) | $\{1, 0.1\} \in c$ | 0.8815 | 0.8779 | 0.6799 | 0.6693 |
| **(P1) FedProxLoD** | Parameter-free | 0.8766 | 0.8766 | 0.6856 | 0.6854 |
| **(P2) FedProxWLoD** | Parameter-free | 0.8771 | **0.8771** | **0.6906** | **0.6887** |
| **(P1') (P1) w/o model merge** | Parameter-free | 0.8766 | 0.8734 | 0.6778 | 0.6722 |
| **(P2') (P2) w/o model merge** | Parameter-free | **0.8779** | 0.8750 | 0.6829 | 0.6751 |

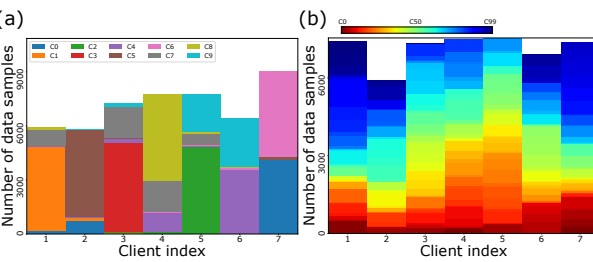

Figure 8: Data distributions using $n = 7$ and $\alpha = 0.1$: (a) fMNIST classification in (T1) and (b) CIFAR-100 classification in (T2).

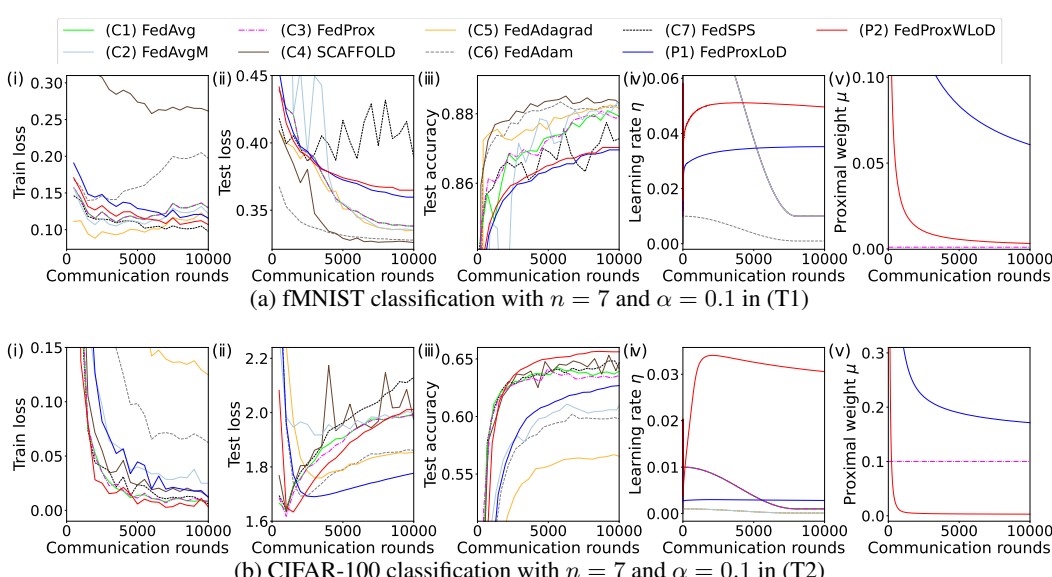

(a) fMNIST classification with $n = 7$ and $\alpha = 0.1$ in (T1)

(b) CIFAR-100 classification with $n = 7$ and $\alpha = 0.1$ in (T2)

Figure 9: Convergence curves illustrating (i) train loss, (ii) test loss, and (iii) test accuracy, and the evolution of (iv) learning rate and (v) proximal weight when $n = 7$ and $\alpha = 0.1$.

Table 5: Best test accuracy under $n = 7$ and $\alpha = 0.1$. In the comparing algorithms (C1)-(C7), pre-tuning of parameters was conducted. Despite not requiring parameter pre-tuning, our parameter-free algorithms (P1), (P2), (P1'), (P2') achieved competing performance relative to the best performance of pre-tuned baseline algorithms (C1)-(C7).

| Algorithms | Parameters | (T1) Convex-fMNIST | | (T2) ResNet-18-CIFAR-100 | |
|---|---|---|---|---|---|
| | | Best test acc. | Last test acc. | Best test acc. | Last test acc. |
| **(C1) FedAvg** McMahan et al. (2017) | $\{1, 0.1, 0.01, 0.001\} \in \eta$ | 0.8809 | 0.8793 | 0.6427 | 0.6393 |
| **(C2) FedAvgM** Hsu et al. (2019) | $\{1, 0.1, 0.01, 0.001\} \in \eta$ | 0.8835 | 0.8815 | 0.6515 | 0.6351 |
| **(C3) FedProx** Li et al. (2020) | $\{1, 0.1, 0.01, 0.001\} \in \eta$ $\{0.1, 0.01, 0.001\} \in \mu$ | 0.8808 | 0.8792 | 0.6426 | 0.6410 |
| **(C4) SCAFFOLD** Karimireddy et al. (2020) | $\{1, 0.1, 0.01, 0.001\} \in \eta$ | **0.8851** | **0.8831** | **0.6532** | **0.6469** |
| **(C5) FedAdaGrad** Reddi et al. (2020) | $\{1, 0.1, 0.01, 0.001\} \in \eta$ | 0.8825 | 0.8814 | 0.6174 | 0.6120 |
| **(C6) FedAdam** Reddi et al. (2020) | $\{1, 0.1, 0.01, 0.001\} \in \eta$ | 0.8834 | 0.8826 | 0.6289 | 0.6171 |
| **(C7) FedSPS** Mukherjee et al. (2023) | $\{1, 0.1\} \in c$ | 0.8775 | 0.8728 | 0.6483 | 0.6417 |
| **(P1) FedProxLoD** | Parameter-free | 0.8696 | 0.8696 | 0.6268 | 0.6268 |
| **(P2) FedProxWLoD** | Parameter-free | 0.8703 | **0.8703** | **0.6566** | **0.6562** |
| **(P1') (P1) w/o model merge** | Parameter-free | 0.8706 | 0.8694 | 0.6259 | 0.6207 |
| **(P2') (P2) w/o model merge** | Parameter-free | **0.8718** | 0.8702 | 0.6397 | 0.6344 |

**Memory usage.** We evaluated the peak memory usage of each algorithm on five communication rounds. The results in Table 6 show that although the proposed methods require somewhat more memory than the baseline algorithms, the difference is not substantial.

Table 6: Peak memory usage within five communication rounds for each benchmark test.

| Algorithms | (T1) Convex-fMNIST | | (T2) ResNet-18-CIFAR-100 | | (T3) BERT-SST-2 | |
|---|---|---|---|---|---|---|
| | RAM (GB) | VRAM (GB) | RAM (GB) | VRAM (GB) | RAM (GB) | VRAM (GB) |
| **(C1) FedAvg** McMahan et al. (2017) | 0.3897 | 0.4830 | 0.1383 | 0.4206 | 0.5882 | 3.1870 |
| **(C2) FedAvgM** Hsu et al. (2019) | 0.3930 | 0.4831 | 0.1847 | 0.4206 | 0.9300 | 3.1863 |
| **(C3) FedProx** Li et al. (2020) | 0.3897 | 0.4863 | 0.1383 | 0.4711 | 0.5882 | 3.5491 |
| **(C4) SCAFFOLD** Karimireddy et al. (2020) | 0.3963 | 0.4929 | 0.2289 | 0.5625 | 1.2696 | 4.2571 |
| **(C5) FedAdaGrad** Reddi et al. (2020) | 0.3964 | 0.4831 | 0.2299 | 0.4206 | 1.2718 | 3.1863 |
| **(C6) FedAdam** Reddi et al. (2020) | 0.3963 | 0.4831 | 0.2299 | 0.42065 | 1.2718 | 3.1862 |
| **(C7) FedSPS** Mukherjee et al. (2023) | 0.3897 | 0.4831 | 0.1383 | 0.4206 | 0.5882 | 3.1869 |
| **(P1) FedProxLoD** | 0.4028 | 0.4864 | 0.3224 | 0.4706 | 1.9663 | 3.5493 |
| **(P2) FedProxWLoD** | 0.4028 | 0.4864 | 0.3224 | 0.4713 | 1.9663 | 3.3902 |
| **(P1') (P1) w/o model merge** | 0.4028 | 0.4864 | 0.3224 | 0.4714 | 1.9663 | 3.3902 |
| **(P2') (P2) w/o model merge** | 0.4028 | 0.4864 | 0.3224 | 0.4711 | 1.9663 | 3.3905 |

## F  LIMITATIONS AND FUTURE WORK

Compared to the original FedProx, our proposed FedProxLoD and FedProxWLoD, described in Algorithm 1, achieve parameter-free FL. However, this is realized at the cost of introducing additional operations: i) global loss computation on the central server (in Line 12), ii) extra model merge motivated by our convergence analysis (in Lines 22–24 in Algorithm 1), and iii) transmission of additional information (in Lines 8 and 25 in Algorithm 1). Among these, iii) is minor, as the additional transmitted quantities are scalars and incur negligible cost. Regarding (ii), we empirically evaluated the impact of the extra model merge through ablation studies in Section 5 and Appendix E. As discussed in Section 5, while this operation is necessary for theoretical convergence guarantees, it does not appear to be critical for empirical performance improvements. In other words, if rigorous convergence guarantees are not required, the extra model merging procedure can be reasonably omitted in practice. Finally, concerning (i), as described in Appendix D, the global loss using global parameter $f(\mathbf{x}^{(t+1)})$ is computed using $1,000$ data samples homogeneously picked from the dataset. This may limit the applicability of our methods in settings where any form of data aggregation on the central server is strictly prohibited.

Furthermore, regarding the limitations of our theoretical contribution (Theorem 2), the convergence rates are estimated for $G$-Lipschitz convex loss functions. This follows the principles in DoG and DoWG, which are also analyzed under convex settings. To enhance practical relevance, we conducted empirical evaluations on both convex and non-convex loss functions, including deep learning models. Additionally, although we implemented the proposed algorithms to allow multiple local parameter updates (in Algorithm 3 in Appendix D), we do not currently provide a theoretical analysis of the approximation gap. Addressing this gap theoretically remains an important direction for future work.

A potential risk of our proposed FedProxLoD and FedProxWLoD is the privacy concern associated with possible information leakage through loss differences. This issue is beyond the scope of the present work, and we leave it for future investigation. Within a Differential Privacy (DP) framework, one could address this by analyzing the sensitivity of the loss differences to a change in a single data sample. While we do not study such leakage in this paper, we note that leakage from high-dimensional gradients is typically more severe than that from scalar loss differences.

## G  IMPACT STATEMENT

We present parameter-free federated learning algorithms, which can be applied for training large-scale models across extensive distributed computing resources, such as data centers. While this approach removes the need for parameter pre-tuning, a potential risk is that it could enable a wider range of organizations to train large models more efficiently.

