# OpenReview forum: "FedProx-based heterogeneity-aware parameter-free federated learning"
_ICLR.cc/2026/Conference — Submitted to ICLR 2026_

### Official Review · Reviewer_cLk6 · 2025-10-31

**Soundness:** 3
**Presentation:** 3
**Contribution:** 2
**Rating:** 4
**Confidence:** 4

**Summary:**

This paper develops a parameter-free method for federated learning focusing on FedProx. It allows dynamic adjustment of learning rates as well as the proximal-term weights based on historical gradients and losses, so that they do not need to be manually tuned. A convergence analysis is provided for convex objectives. Experiments on three datasets are conducted to showcase the competitive performance of the proposed approach compared to STOA FL algorithms.

**Strengths:**

- The parameter-free approach is well motivated for federated learning, which typically requires expensive training and hyper-parameter tuning.

- The reviewer took a quick read over the proofs and the convergence result looks solid. Importantly, it matches the order with centralized learning.

- The paper is very well written and easy to read.

**Weaknesses:**

- Assumption 6 looks problematic. When clients have heterogeneous data distributions, how can one assume there is a shared minimizer for each client's local objective? Can the author clarify on this or is this just a typo?  Also, in line 118, what does monotonic decrease mean? Is it with respect to time slot t? If so, this is a very strong assumption stating that each step the model is shifting closer to the minimizer, which is unlikely to hold. Let's take a step back and say this is true when one very lucky, but wouldn't this be a result instead of an assumption?

- My major concern with the parameter-free approach (e.g., in (2)) is that, while it saves some computation on hyper-parameter tuning, it requires storage of all historical models/gradients in order to compute the hyper-parameters. This tradeoff needs to be specifically discussed and evaluated in the paper, e..g., one could use a table comparing memory usage across all benchmarks.

- In Lemma 2, \Delta is function of \mu, but at the same time, in (6), \mu is a function of \Delta. Can the authors provide explicit formulas for both terms for clarify?

- The reviewer is curious why this paper focuses on FedProx. The parameter-free approach seems to naturally apply to other FL approaches such as FedAvg and FedOpt. Is there a reason for building the approach based off FedProx?

- For the experiment on FMNSIT, why pick a half-frozen MLP? How is the other layer (including activation function) designed? If the authors intended to use a convex model, why not use something like logistic regression?

- Another major concern pertains to the current experiment results. In Table 2, all methods achieve very similar performance on the three datasets.  If we further look at Fig. 2aiii and 2ciii, the proposed method achieves the worst performance compared to baselines. While this might be due to a mild data heterogeneity, but if we look at Table 2 and Fig.3 with a larger heterogeneity, the proposed method falls behind with a larger gap to baselines. It would be more meaningful to showcase the computation saving  (as well as the memory usage) of the proposed method.

**Questions:**

See above.

---

> ### Author Response · Authors · 2025-11-20
>
> # [W1] Concerns regarding Assumption 6 and convex-model experiments.
>
> > Assumption 6 looks problematic [...].
>
> > For the experiment on FMNIST, why pick a half-frozen MLP? [...]
>
> As these two weaknesses are interrelated, we address them together.
>
> First, Assumption 6 corresponds to the interpolation regime, i.e., training loss $f_i$ can be zero, which typically holds for overparameterized models. In this context, we therefore require an overparameterized model (with large $m$). Accordingly, we require a sufficiently overparameterized model (large $m$) and therefore use a half-frozen MLP rather than a convex model such as logistic regression. We explained this context at lines 118-120 in Section 2 and lines 374–375 in Section 5.
>
> # [W2] Misunderstanding in historical models/gradients storage.
>
> > [...] it requires storage of all historical models/gradients in order to compute the hyper-parameters. [...]
>
> Regarding your comment that "it requires storage of all historical models/gradients", this is NOT true. We may have failed to make this clear. Could you point us to the specific sentences/phrases that gave this comment? We will revise the presentation accordingly.
>
> Given your concerns about memory usage, we evaluated it on three benchmarks. Using logs from the first 5 communication rounds, we summarize the results as follows:
>
> **[Peak memory usage in FedProxWLoD with three benchmark tests]**
>
> Left value in each cell: main memory (RAM), in GB,  right value in each cell: GPU memory (VRAM), in GB.
>
> **(T1) Convex-fMNIST classification $(\alpha = 1)$ for 5 communication rounds**
>
> | Communication round|r=1|r=2|r=3|r=4|r=5|
> |-------------| --- | --- | --- | --- | --- |
> |(P2) FedProxWDoL | 47.7997/0.0487 | 47.8001/0.0487 | 47.8005/0.0487 | 47.8005/0.0487 | 49.8005/0.4897 |
>
> **(T2) ResNet-18-CIFAR-100 classification $(\alpha = 1)$ for 5 communication rounds**
>
> | Communication round|r=1|r=2|r=3|r=4|r=5|
> |-------------| --- | --- | --- | --- | --- |
> |(P2) FedProxWDoL | 49.4779/0.4873 | 49.4780/0.4897 | 49.5655/0.4897 | 49.4780/0.4897 | 49.4825/0.4897 |
>
> **(T3) BERT-SST-2 classification $(\alpha = 1)$ for 5 communication rounds**
>
> | Communication round|r=1|r=2|r=3|r=4|r=5|
> |-------------| --- | --- | --- | --- | --- |
> |(P2) FedProxWDoL | 51.3129/3.7083 | 51.3129/3.7083 | 51.3129/3.7083 | 51.3129/3.7083 | 51.3129/3.7083 |
>
> # [W3] Misunderstanding in relationship between $\Delta$ and $\mu$.
>
> > In Lemma 2, $\Delta$ is function of $\mu$, but at the same time, in (6), $\mu$ is a function of $\Delta$. [...]
>
> Lemma 2 might lead to your misunderstanding that enables simultaneous conversion between $\Delta$ and $\mu$. In Algorithm 1 (lines 12, 15, 19), $\Delta$ and $\mu$ are alternately updated in a recurrent manner, not jointly. Our convergence analysis (yielding Theorem 2) is carried out for Algorithm 1 under this alternating update scheme. If you have suggestions on how to prevent this misunderstanding, we would appreciate them and will revise accordingly.
>
> # [W4] Why begin with FedProx?
>
> > The reviewer is curious why this paper focuses on FedProx. [...]
>
> This is a good question. (A similar question comes from another reviewer gYee. )
>
> First, our interest lies in parameter-free optimization. Among many approaches, DoG and DoWG have performed well empirically even with non-convex DNNs; our main idea is to leverage these successes to Federated Learning (FL).
>
> Within this context, we examine which fundamental FL algorithms (e.g., FedAvg, FedProx) yield an inequality analogous to Lemma 1-the key lemma underlying DoG/DoWG. We find that this holds only for FedProx, which leads to Lemma 2. Upon reconsideration, we conclude that Lemma 1 mirrors the descent inequality of the classical proximal-point method, which FedProx implicitly includes.
>
> Although parts of this are noted at lines 075–077 and 212–213, we have added the motivation for starting with FedProx in Appendix C.
>
> # [W5] Concerns in experiments.
>
> > [...] In Table 2, all methods achieve very similar performance on the three datasets. [...]
>
> You might expect parameter-free FL algorithms to surpass the baselines. However, as noted in lines 339-341, "the main goal is to assess whether the proposed parameter-free FL algorithms can achieve comparable to, or better than, baseline FL algorithms with pre-tuned parameters". Figures 2 and Table 2 are consistent with this aim. While you may have concerns in Figures 2(a) and 2(c), those results are comparable to the corresponding parameter-tuned baselines.
>
> > It would be more meaningful to showcase the computation saving (as well as the memory usage) of the proposed method.
>
> During the rebuttal phase, we evaluated the memory usage of each algorithm (see Table 6 in Appendix E). This table shows that although the proposed method uses somewhat more memory than the baseline algorithms, the difference is not substantial.

---

> > ### Comment · Reviewer_cLk6 · 2025-11-25
> > **Acknowledging the rebuttal**
> >
> > I appreciate the authors' detailed response. Most of my concerns/confusions have been cleared, except for Assumption 6. Although it holds in an interpolation regime with a common minimizer across clients, this is still restrictive for heterogeneous FL. Can the authors use experiments to validate this assumption as well as the associated monotonic decrease of the term in line 125?

---

> ### Author Response · Authors · 2025-11-29
>
> We are glad to hear that most of your concerns have been mitigated. In response to your additional questions regarding Assumption 6, we conducted an extra experiment. Using the (T3) BERT-SST-2 task, we trained a model on a single node with the full dataset for a sufficiently large number of iterations. The resulting model can be regarded as $x*$. We then computed the global objective $f(x*)=0.0345$ and the average local objective $\sum_{i=1}^{n} f_i(x*)=0.026$. Since both values are close to zero, we believe this provides empirical support for Assumption 6.
>
> > as well as the associated monotonic decrease of the term in line 125?
>
> Because the local model updates are based on stochastic gradients, we cannot observe a strictly monotonic decrease numerically. However, this is not inconsistent with our theoretical results.

---

### Official Review · Reviewer_DPv8 · 2025-11-01

**Soundness:** 4
**Presentation:** 4
**Contribution:** 3
**Rating:** 4
**Confidence:** 4

**Summary:**

This paper adapts the DoG/DoWG parameter-free optimization methods to the federated setting by combining them with FedProx. Specifically, the proximal weight $\mu$ in FedProx is set to be the square root of the sum of the client losses divided by how far the global model is from the start. The clients solve their client subproblems by using DoG, which then makes the entire algorithm learning-rate free and makes it so $\mu$ does not need to be tuned. Because the method is able to adapt to the heterogeneity of the federated learning problem, it has a heterogeneity-dependent learning rate which demonstrates that it can improve as heterogeneity decreases.

**Strengths:**

- The method makes a thoughtful adaptation of the DoG/DoWG-style parameter-free optimization methods to the federated setting. They are able to rigorously define the analogous quantities relevant to setting $\mu$, and take care to ensure the communication cost is not too large as a result
- They are able to derive a convergence rate depending on heterogeneity, which provides an option for optimization when problems are not too heterogeneous
- Experimental results are fair and trustable, with good tuning of baselines. They are able to show that their algorithm is able to outperform other parameter-free methods and achieve results comparable to the best empirical result from tuned methods
- The method does not require manual tuning, which is great
- Highly readable paper, very well written

**Weaknesses:**

- There are other works that have derived heterogeneity-dependent convergence rates in FL. Should probably compare against them. For example one notable one is below but please survey the literature more thoroughly:

Woodworth, Blake E., Kumar Kshitij Patel, and Nati Srebro. "Minibatch vs local sgd for heterogeneous distributed learning." Advances in Neural Information Processing Systems 33 (2020): 6281-6292.

- I am unconvinced about the setting for the paper. FL and on-device learning has evolved significantly since the original FedAvg paper was published, but the paper does not seem to have kept up with the developments. It is still working on assumptions and settings that were relevant 5 years ago. For example, FL deployments today rely on pretraining, LLMs, or even a combination of both, but the paper does not mention this or compare against it. Pretraining in particular is critical because it is known that it can help avoid the harmful effects of heterogeneity. See the following works

Nguyen, John, et al. "Where to begin? on the impact of pre-training and initialization in federated learning." arXiv preprint arXiv:2206.15387 (2022).

Hou, Charlie, et al. "Private federated learning using preference-optimized synthetic data." arXiv preprint arXiv:2504.16438 (2025).

Wu, Shanshan, et al. "Prompt public large language models to synthesize data for private on-device applications." arXiv preprint arXiv:2404.04360 (2024).

- Is communicating loss differences acceptable from a deployment perspective (privacy)? Again I think the authors should more carefully examine the motivation for the setting they study. The paper does not consider privacy and I understand it is out of the scope of the work, but you need to convince the reader that the setting is practical somehow.
- Are there speedruns (or similar) in the FL optimization community? Without them it is hard to determine whether proposed algorithms are actual improvements. See https://kellerjordan.github.io/posts/speedrun/

**Questions:**

See weaknesses above

---

> ### Author Response · Authors · 2025-11-20
>
> # [W1] Other heterogeneity-dependent convergence rates in FL.
>
> > There are other works that have derived heterogeneity-dependent convergence rates in FL. [...]
>
> Our main claim is that we proposed heterogeneity-aware "parameter-free" federated learning algorithms. Since our convergence analysis is based on the $G$-Lipschitz assumption (Assumption 4), non-smooth (general) convex functions are available as $f_i$.
>
> By contrast, the reference you suggested shows heterogeneity-dependent convergence rates in FL (not parameter-free). In addition, its analysis is performed under $L$-smooth assumption ($f_i$ is required to be smooth). This distinction is the critical difference between our work and your suggested reference.
>
> To make this point explicit, we have revised sentences and added your suggested reference in the Introduction.
>
> # [W2] Relevant works.
>
> > FL and on-device learning have evolved significantly since the original FedAvg paper was published, but the paper does not seem to have kept up with the developments. [...]
>
> Thank you for highlighting recent trends in practical FL deployments. Although we have not yet evaluated LLM pre-training (left for future work), Section 5 already includes (T2) pre-training for image classification on CIFAR-100 with a non-convex model (ResNet-18) and (T3) fine-tuning a non-convex NLP model (BERT) on SST-2. As parameter-free optimization in FL has only been studied in the past few years, it is natural to begin with well-used benchmark tests.
>
> # [W3] Privacy concerns.
>
> > Is communicating loss differences acceptable from a deployment perspective (privacy)?
>
> As you noted, privacy concerns about potential leakage from loss differences are out of scope for this work and left to future study (see Appendix F). Within a DP framework, one could address this by analyzing the sensitivity of loss differences to a single data sample change. While we have not analyzed leakage from loss differences in this paper, we note that leakage from high-dimensional gradients is typically more severe than from scalar loss differences.
>
> # [W4] Evaluation metric.
>
> > Are there speedruns (or similar) in the FL optimization community?
>
> In the rebuttal phase, we evaluated the memory usage of each algorithm on five communication rounds (see Table 6 in Appendix E). The results show that although the proposed methods require somewhat more memory than the baseline algorithms, the difference is not substantial. We hope this mitigates your concern.

---

> ### Author Response · Authors · 2025-11-29
>
> Dear Reviewer DPv8,
>
> We would like to follow up regarding our rebuttal respectfully. We understand that you have a significant workload during the review period, but we would greatly appreciate it if you could re-evaluate our revised paper and rebuttal comments.
>
> We would be grateful for any further comments or clarifications you may have. Your feedback is extremely valuable to us.
>
> Thank you very much for your precious time and consideration.
>
> Best regards,
> Authors

---

### Official Review · Reviewer_gYee · 2025-11-05

**Soundness:** 1
**Presentation:** 3
**Contribution:** 2
**Rating:** 2
**Confidence:** 4

**Summary:**

The paper introduces two "parameter-free" FL algorithms, FedProxLoD and FedProxWLoD, inspired by two algorithms from the centralized setting: DoG and DoWG. By noticing a structural similarity between the convergence analyses of DoG/DoWG and the Proximal Point Algorithms, the paper suggests using adaptive proximal weights based on adaptive learning rates for Stochastic Gradient Descent and Gradient Descent as described in DoG and DoWG. Theoretically, the analysis claims to show that as heterogeneity diminishes, the algorithms offer improved convergence guarantees compared to DoG and DoWG. Numerical results show that the proposed "parameter-free" FedProx-based algorithms perform comparably to the tuned baseline algorithms under moderate and low heterogeneity.

**Strengths:**

The algorithm adopts the adaptive learning rate schemes to develop parameter-free FL algorithms with different aggregation schemes to ensure convergence.

The paper presents mildly non-conventional contributions. It incorporates the concept of "parameter-free" algorithms using DoG and DoWG algorithms in FL. The paper includes adaptive schemes for the proximal coefficient in FedProx and formulates the algorithm to ensure convergence by introducing double merging and calculating the loss difference.

The paper is well-written, with clearly stated assumptions, detailed notations and results, and a well-articulated motivation for the research. The theorems and results were systematically laid out.

**Weaknesses:**

**Justification for poor soundness: major error; see below**

The bound in Lemma 10 is derived as $K$ approaches infinity, but we do not consider this in the non-asymptotic analysis. Intuitively, if $K$ were to approach infinity, the client's local update would drift too far from the global update, causing the client's drift to become unbounded and preventing convergence. Therefore, the bound on the difference between server and client updates in Lemma 10 should be re-examined, as they influence the overall convergence analysis. Additionally, there is no guarantee that $\eta^\prime\leq \frac{1}{\mu^(t)}$ is sufficiently small.

**The submission has limited significance**

In practice, to compute $\Delta^{(t+1)}$ in Line 12, $f(x^{(t+1)})$ cannot be computed on the server, as data cannot be stored there; therefore, the routine outlined in Algorithm 4 is the only option. Including experiments based on Algorithm 4 would thus provide a clearer view of the empirical performance of the proposed algorithms. Comparing with baseline algorithms is unfair if the data is stored on the
server to compute $f(x^{(t+1)})$. Furthermore, to implement FedProx for deep neural networks, SGD-based updates are performed on
the clients for K rounds, as shown in Algorithm 3. This approach employs a fixed learning rate for one communication round, repeated 100 times per client, which can be relatively large, as it is determined by the server based on collective client information. How does the performance of the algorithm vary when a decaying learning rate (e.g., using cosine annealing) is used at the clients as
opposed to the learning rate in Algorithm 3? In that case, does this "parameter-free" method for the learning rate $\eta$ along with proximal coefficient $\mu^{(t)}$ offer any benefits over a tuned learning rate at the clients?

Finally, although the centralized algorithms use the term "parameter-free", the reviewer would like to suggest that there is no reason to carry it forward; how about calling it just "hyper-parameter free"?

**Questions:**

The facts in Table 1 are incorrect. FedAvg (Karimireddy et al., 2020) is analyzed assuming a smooth function and $(G-B)$ bounded gradient dissimilarity, rather than under the assumption of a $G$-Lipschitz function. The $(G-B)$ bounded gradient dissimilarity is also a measure of heterogeneity. Similarly, FedProx (Li et al., 2020) also employs the $B$-bounded dissimilarity assumption. The claim that only the proposed algorithms have a heterogeneity-aware convergence rate is unjustified. Accordingly, the derived convergence rates for FedProxLoD and FedProxWLoD are based on stronger conditions of a $G$-Lipschitz function compared to existing heterogeneity-aware FL algorithms, which are derived under the $L$-smoothness assumption. Additionally, assumption 6, which states that there exists a consistent minimiser $x^*$ for each client $f_i$, is essentially the interpolation condition discussed in the paper and is a stronger assumption. Thus, the given convergence rates are under stronger conditions of- 1) $G$-Lipschitz function, 2) interpolation condition, and 3) full gradient computation and full client participation. Based on these assumptions, the convergence rates are not tighter than those of existing rates. Could you please comment on this?

The DoG algorithm has a high probability convergence bound for the optimality gap $f(x_T)-f(x^*)$, and DoWG has deterministic convergence rates for GD with the learning rate scheme provided by DoWG; however, Theorem 1 combines these two different notions of convergence and presents a single theorem for both, which is again not factually correct. Lemma 2 shares a parallel structure with Lemma 1 and provides motivation for using DoG/DoWG-inspired schemes for the adaptive proximal coefficient. However, the similarity feels forced, as $\mu^(t)$ is multiplied and divided by the second term on the right-hand side to create a structurally similar lemma. If, instead, the right-hand side is minimized over $\mu^(t)$ without explicitly including it in the second term, and an adaptive formulation of \mu^(t) is based on this, how would that affect convergence?

Moreover, in classical SGD and even in FL, $\eta$ is tuned for improved convergence guarantees. DoWG and DoG utilize the gradient norm squared and the distance to the initial point, based on the effective step size in Normalised GD, for optimal performance. What is the intuition behind using an adaptively constructed proximal coefficient?

In line 395, there is a minor typo; it is stated, “We use a batch size of 64, and examine three levels of data heterogeneity.” However, the experiments are included only for two values of $\alpha$, i.e., $\alpha=1,0.1$.

Appendix A reproduces the proofs of DoWG without proper citations, making them redundant. The paper should include only essential parts and direct readers to find detailed proofs in the DoWG paper. The equation on line 929 is incorrect. It should be $\nabla f_i(x_i^t)+\frac{1}{\eta'}(y - x_i^t)+\mu(yx^t) = 0$. Expression in line 1165 is incorrect and should be omitted.

**REPRDOCUBILITY**: Insufficient amount of details available. Could you please justify that the information provided in the paper are sufficient for the reproducibility of the results?

---

> ### Author Response · Authors · 2025-11-20
>
> # [W1] Misunderstanding in Lemma 10.
> > major error. The bound in Lemma 10 is derived as $K$ approaches infinity, but we do not consider this in the non-asymptotic analysis. [...]
>
> Thank you very much for your important comment. We respectfully clarify that there is no error in the statement or the proof of Lemma 10.
>
> First, we emphasize that the convergence analysis (Theorem 2) is performed for Algorithm 1, where the approximated local update rule is not used (see line 6). We suspect your confusion may arise from this distinction. As stated in lines 322–323, extending the analysis to fully incorporate the effect of the approximate local update is an important direction, and we clearly leave it for future work.
>
> If you had already noticed this distinction but still raised the concern that "_as $K$ were to approach infinity, the client's local update would drift too far from the global update_", we would like to clarify that this situation does not occur under $\mu$-strongly convex regularized local subproblem. In such cases, each regularized local objective has a unique minimizer, and any reasonable optimizer (with any properly chosen learning rate) will converge to it. Hence, due to the regularization term $\frac{\mu^{(t)}}{2} || x - x_\mathrm{best}^{(t)} ||^2$, the client drift remains bounded, exactly as stated in Lemma 10. This fact also plays an important role in the FedProx paper.
>
> Furthermore, to analyze the solution to line 6 of Algorithm 1, we examine the infinite-iteration limit of that subproblem by introducing a conceptual learning rate $\eta^{\prime}$. This learning rate is used solely for characterizing the unique minimizer of the subproblem; it is unrelated to the approximate $K$ steps local updates with learning rate $\eta^{(t)}$ in Algorithms 3 and 4. To avoid potential confusion (e.g., $k$ used for the limit analysis resembles $K$), we have revised the notation in Lemma 10 accordingly.
>
> We hope this addresses your concern and helps dispel any misunderstanding.
>
> # [W2] Implementation concerns in the numerical experiments.
> > [...] The submission has limited significance.  [...] the routine outlined in Algorithm 4 is the only option. [...]
>
> Thank you for raising this practical concern. We agree that, in federated learning settings, the server cannot access or store local data, and therefore cannot directly compute $f(x^{(t)})$ in Line 12. However, we would like to clarify that this does not affect either the correctness or the fairness of our empirical evaluation.
>
> Importantly, the computation in Line 12 of Algorithm 1 is not essential to the algorithm itself. It can always be replaced by the procedure in Lines 17–22 of Algorithm 4, which evaluates the objective value using clients’ local data without requiring the server to store any data. This replacement preserves the federated setting.
>
> Moreover, Lines 17–22 of Algorithm 4 are mathematically equivalent to Line 12 of Algorithm 1, except for the additional communication required to send the aggregated model $x^{(t)}$ to clients. The communication cost of sending scalar objective values back to the server is negligible. Therefore, using Lines 17–22 of Algorithm 4 instead of Line 12 of Algorithm 1 yields exactly the same algorithmic behavior and does not change any theoretical properties.
>
> Consequently, Algorithm 1—after simply replacing Line 12 with Lines 17–22 of Algorithm 4—remains fully consistent with all theoretical guarantees presented in the paper. This also implies that our experimental comparisons remain fair under the standard federated protocol, because Algorithm 3 (used in our experiments) and Algorithm 4 are mathematically equivalent.
>
> We hope this clarifies the concern.
>
> > [...] How does the performance of the algorithm vary when a decaying learning rate (e.g., using cosine annealing) is used at the clients as opposed to the learning rate in Algorithm 3? [...]
>
> For this question, we will try to report additional results if feasible.
>
> # [W3] Concerns in terminology "parameter-free".
> > [...] how about calling it just "hyper-parameter free"?
>
> We partially agree; however, in this field, optimization that makes hyperparameter tuning free has traditionally been referred to as parameter-free optimization.

---

> ### Author Response · Authors · 2025-11-20
>
> # [Q1] Concerns in Table 1.
>
> > The facts in Table 1 are incorrect. FedAvg (Karimireddy et al., 2020) is analyzed assuming a smooth function and $(G, B)$ bounded gradient dissimilarity, rather than under the assumption of a $G$-Lipschitz function.
>
> Following your feedback, we have completely revised Table 1. In the revised paper, the convergence rates of FedAvg and FedProx are now heterogeneity-aware under $(\zeta, B)$-Bounded Gradient Dissimilarity (BGD) assumption. If you still have any concerns about Table 1 in the revised paper, we would be grateful if you could share them with us.
>
> In addition, we investigated the relationship between $(\zeta, B)$-BGD and our $\zeta$-bounded data heterogeneity (Assumption 5). As proved in Appendix C.2, if Assumption 5 holds, then $(\sqrt{2} \zeta, \sqrt{2})$-BGD holds. We thank you for providing us with the opportunity to examine this important relationship.
>
> We are glad if your concern is mitigated.
>
> > The claim that only the proposed algorithms have a heterogeneity-aware convergence rate is unjustified.
>
> Our claim is that we proposed heterogeneity-aware **parameter-free** federated learning algorithms. To make this explicit, several phrases in the Introduction are revised.
>
> > Accordingly, the derived convergence rates for FedProxLoD and FedProxWLoD are based on stronger conditions of a $G$-Lipschitz function compared to existing heterogeneity-aware FL algorithms, which are derived under the $L$-smoothness assumption.
>
> Although you may believe that the $G$-Lipschitz function assumption is stronger than the $L$-smooth assumption, this is NOT true. More precisely, these two assumptions are incomparable, and we do not think one can be categorically stated to be weaker or stronger than the other. For example, non-smooth functions are allowed under the $G$-Lipschitz assumption, whereas $L$-smoothness requires differentiability and a Lipschitz-continuous gradient.
>
> > 2) [...] interpolation condition,
>
> As you pointed out, Assumption 6 corresponds to the interpolation regime, i.e., training loss $f_i$ can be zero, which typically holds for overparameterized models. In this sense, our setting indeed requires an overparameterized model (with large $m$).
>
> Although this may be a strong assumption, as you noted, it is necessary for our rigorous convergence analysis. To make this explicit, we have added a column in Table 1 indicating whether the interpolation condition is required.
>
> > 3) full gradient computation and full client participation.
>
> As you pointed out, our analysis is based on full gradient computation and full client participation, and its expansion to mitigate these requirements remains future work. At least, in our implementation used in numerical experiments, we use stochastic gradients.
>
> > Based on these assumptions, the convergence rates are not tighter than those of existing rates.
>
> Based on these assumptions, we establish tight convergence rates for our parameter-free federated learning algorithms.
>
> # [Q2] Concerns in Theorem 1.
>
> >The DoG algorithm has a high probability convergence bound [...]
>
> Thank you for pointing this out. This is correct: Appendix A is intended to establish that DoG possesses a deterministic convergence rate for the optimality gap $f(x^{(T)}) - f(x^{\ast})$. To articulate this precisely, we have added the explanations accompanying Theorem 1 and clarified the purpose of Appendix A.
>
> # [Q3] Concerns in Lemma 2.
>
> > Lemma 2 shares a parallel structure with Lemma 1 and provides motivation for using DoG/DoWG-inspired schemes for the adaptive proximal coefficient. However, the similarity feels forced [...].
>
> We strongly disagree with your opinion that _"the similarity feels forced_". To leverage the techniques used from Lemma 1 to Theorem 1, we derived Lemma 2 to be structurally analogous to Lemma 1. Moreover, even without explicitly placing $\mu^{(t)}$ in the second term, the proof necessarily generates an implicit decomposition in $\mu^{(t)}$, yielding an equivalent conclusion.
>
> # [Q4] Why begin with FedProx (with adaptive proximal coefficient)?
>
> > [...] What is the intuition behind using an adaptively constructed proximal coefficient?
>
> This is a good question. (A similar question comes from reviewer cLk6. )
>
> First, our interest lies in parameter-free optimization. Among many approaches, DoG and DoWG have performed well empirically even with non-convex DNNs; our main idea is to leverage these successes to Federated Learning (FL).
>
> Within this context, we examine which fundamental FL algorithms (e.g., FedAvg, FedProx) yield an inequality analogous to Lemma 1-the key lemma underlying DoG/DoWG. We find that this holds only for FedProx, which leads to Lemma 2. Upon reconsideration, we conclude that Lemma 1 mirrors the descent inequality of the classical proximal-point method, which FedProx implicitly includes.
>
> Although parts of this are noted at lines 075–077 and 212–213, we have added the motivation for starting with FedProx in Appendix C.

---

> ### Author Response · Authors · 2025-11-20
>
> # [Q5] Typos regarding $\alpha$-selection.
>
> > In line 395, there is a minor typo; [...]
>
> Revised. Thank you for pointing this out.
>
> # [Q6] Redundant proof in Appendix A.
>
> > Appendix A reproduces the proofs of DoWG without proper citations, making them redundant. [...]
>
> A motivation for including Appendix A is to help readers clearly and efficiently understand the differences between the convergence analysis in DoG/DoWG and our FedProxLoD/FedProxLoWD. Accordingly, we have added Appendix A.1 to clarify the positioning of this section with citations.
>
> > The equation on line 929 is incorrect.  Expression in line 1165 is incorrect and should be omitted.
>
> Revised. Thank you for the helpful suggestions.
>
> # [Q7] Concerns in reproducibility.
>
> >[...] Could you please justify that the information provided in the paper are sufficient for the reproducibility of the results?
>
> We provided the source code used in experiments and summarized associated hyperparameter sets and computing resources in the main paper and Appendix E. Could you please let us know what additional details would help with reproduction?

---

> ### Author Response · Authors · 2025-11-29
>
> Dear Reviewer gYee,
>
> We would like to follow up regarding our rebuttal respectfully. We understand that you have a significant workload during the review period, but we would greatly appreciate it if you could re-evaluate our revised paper and rebuttal comments.
>
> We would be grateful for any further comments or clarifications you may have. Your feedback is extremely valuable to us.
>
> Thank you very much for your precious time and consideration.
>
> Best regards,
> Authors

---

### Author Response · Authors · 2025-11-20

Thank you very much for taking the time to review our paper and for providing valuable feedback. We have carefully revised the paper reflecting your comments. In the revised version, **all changes are highlighted in red**.

While we are deeply grateful for your comments, we also felt that several points may stem from misunderstandings or are difficult for us to fully agree with. **We would greatly appreciate it if you could read our rebuttal carefully and kindly reconsider your rating and other evaluations** in light of our responses.

Thank you in advance for your time and consideration.

---

### Meta-Review · Area_Chair_LrdG · 2026-01-07

**Summary:**

Reviewers agree that the paper proposes a novel and technically nontrivial extension of parameter-free optimization (DoG/DoWG) to the federated learning (FL) setting by leveraging FedProx with adaptive proximal coefficients. However, the concerns regarding the assumptions (raised by **Reviewer gYee** and **Reviewers cLk6**, including the interpolation condition, full gradient computation, and full client participation) were only partially addressed. In addition, the concern raised by **Reviewer DPv8** regarding outdated FL setting (with limited discussion of pretraining, LLMs, FL deployments) also remains only partially resolved.

**Reviewer Concerns:**

**Reviewer gYee’s concerns:**

(1) The concerns regarding the soundness of Lemma 10 and the convergence analysis, as well as the realism of the experimental implementation in federated learning settings, appear to have been addressed. In particular, the authors rewrote Lemma 10 and proposed replacing Line 12 of Algorithm 1 with Lines 17–22 of Algorithm 4. However, these changes do not seem to be fully reflected in the current revised version, and this point therefore remains somewhat unclear.

(2) The concerns regarding incorrect or misleading comparisons in Table 1 was adequately addressed by revising Table 1: the authors corrected the assumptions and explicitly indicated the interpolation and heterogeneity conditions under which each method applies.

(3) The concerns regarding the assumptions (interpolation condition, full gradient computation and full client participation) was only partially addressed, as the authors said that their analysis is based on full gradient computation and full client participation, and its expansion to mitigate these requirements remains future work.

**Reviewer DPv8’s concerns:**

(1) The concerns regarding limited comparison with other heterogeneity-dependent FL methods and evaluation metrics were addressed by clarifying the key distinctions between the proposed method and existing heterogeneity-dependent FL approaches, and by adding an explicit memory-usage evaluation.

(2) The main concerns regarding outdated FL setting (limited discussion of pretraining, LLMs, FL deployments) was partially addressed. Section 5 already includes (T2) pre-training for image classification on CIFAR-100 with a non-convex model (ResNet-18) and (T3) fine-tuning a non-convex NLP model (BERT) on SST-2. At the same time, the authors explicitly stated that they have not yet evaluated LLM pre-training (left for future work).

(3) Privacy related issues remain a concern, however, they are acknowledged by Reviewer DPv8 as being outside the scope of the current work.

**Reviewer cLk6’s concerns:**

She/he engaged actively with the authors during the rebuttal and raised concerns regarding Assumption 6 (the interpolation condition), memory overhead, the relationship between \Delta and \mu, the choice of FedProx over FedAvg/FedOpt, and the experimental results in Table 2. After the authors clarified these points and added additional memory-usage experiments, Reviewer cLk6 explicitly stated that most of her/his concerns/confusions have been cleared.

The remaining outstanding concern relates to the use of Assumption 6 (interpolation condition) in heterogeneous federated learning settings and the associated monotonic decrease results, which remain only partially addressed.

**Reviewer Scores:**

**Reviewer gYee** may increase the score (**likely from 2 to 4**), as the main concerns regarding the soundness of Lemma 10, the convergence analysis, and the realism of the experimental implementation in federated learning settings appear to have been addressed.

**Reviewer DPv8** is likely to maintain a score of **4**, since the main concerns regarding outdated FL setting (limited discussion of pretraining, LLMs, FL deployments) was only partially addressed.

**Reviewer cLk6** may maintain a score of **4** or increase it (**likely from 4 to 6**), as several concerns were clarified and partially resolved, while some issues remain outstanding.

---

### Decision · Program_Chairs · 2026-01-26

Reject